



# A global re-analysis of regionally resolved emissions and atmospheric mole fractions of SF$_6$ for the period 2005-2021

Martin Vojta[1], Andreas Plach[1], Saurabh Annadate[2,3], Sunyong Park[4], Gawon Lee[4], Pallav Purohit[6], Florian Lindl[6], Xin Lan[8], Jens Mühle[7], Rona L. Thompson[5], and Andreas Stohl[1]

[1]Department of Meteorology and Geophysics, University of Vienna, Vienna, Austria
[2]Department of Pure and Applied Sciences, University of Urbino "Carlo Bo", Urbino, Italy
[3]University School for Advanced Studies IUSS, 27100 Pavia, Italy
[4]Kyungpook National University, Department of Oceanography, School of Earth System Sciences, Daegu, South Korea
[5]NILU, Kjeller, Norway
[6]International Institute for Applied Systems Analysis (IIASA), Laxenburg, Austria
[7]Scripps Institution of Oceanography, University of California, San Diego, CA, 92037, USA
[8]NOAA Global Monitoring Laboratory, Boulder, CO, USA

**Correspondence:** Martin Vojta (martin.vojta@univie.ac.at)

**Abstract.** We determine the global emission distribution of the potent greenhouse gas sulfur hexafluoride (SF$_6$) for the period 2005-2021 using inverse modeling. The inversion is based on 50-day backward simulations with the Lagrangian Particle Dispersion Model (LPDM) FLEXPART and on a comprehensive observation data set of SF$_6$ mole fractions, in which we combine continuous with flask measurements sampled at fixed surface locations, and observations from aircraft and ship campaigns. We use a global distribution-based (GDB) approach to determine baseline mole fractions directly from global SF$_6$ mole fraction fields at the termination points of the backward trajectories. We compute these fields by performing an atmospheric SF$_6$ re-analysis, assimilating global SF$_6$ observations into modeled global three-dimensional mole fraction fields. Our inversion results are in excellent agreement with several regional inversion studies in the USA, Europe, and China. We find that (1) annual U.S. SF$_6$ emissions strongly decreased from 1.25 Gg in 2005 to 0.48 Gg in 2021, however, they were on average twice as high as the reported emissions to the United Nations. (2) SF$_6$ emissions from EU countries show an average decreasing trend of -0.006 Gg/yr during the period 2005 to 2021, including a substantial drop in 2018. This drop is likely a direct result of the EU's F-gas regulation 517/2014, which bans the use of SF$_6$ for recycling magnesium die-casting alloys from 2018 and requires leak detection systems for electrical switch gear. (3) Chinese annual emissions grew from 1.28 Gg in 2005 to 5.16 Gg in 2021, with a trend of 0.21 Gg/yr, which is even higher than the average global total emission trend of 0.20 Gg/yr. (4) National reports for the USA, Europe, and China all underestimated their SF$_6$ emissions. (5) The global total SF$_6$ emissions are captured well by the inversion, however, results are sensitive to the *a priori* emission estimates, given that substantial biases of these estimates in regions poorly covered by the measurement network (e.g. Africa, South America) can be improved but not entirely corrected. (6) Monthly inversions indicate that SF$_6$ emissions in the Northern Hemisphere were on average higher in summer than in winter throughout the study period.



## 1 Introduction

Sulfur hexafluoride ($SF_6$) is the greenhouse gas (GHG) with the highest known Global Warming Potential (GWP), 24,300, over a 100-year time horizon (Smith et al., 2021). However, this GWP-100 value might still underplay the climate impact of this gas. Once emitted, $SF_6$ accumulates in the atmosphere, as it is only slowly degraded via photolysis and electron attachment (Ravishankara et al., 1993) resulting in a very long atmospheric lifetime, with estimates ranging from 580 to 3200 years (Kovács et al., 2017; Patra et al., 1997; Ravishankara et al., 1993; Ray et al., 2017). The ocean also acts as a sink for atmospheric $SF_6$, however, its magnitude is debated, with estimates ranging up to 7% of the global annual emissions (Ni et al., 2023). Regardless of its exact lifetime and possible ocean sink, $SF_6$ emissions will cause a positive radiative forcing for hundreds of years. Thus, GWPs, which are typically given for time horizons of 20 or 100 years, underestimate the climate impact of $SF_6$ on longer time scales.

Since the early 2000s, global concentrations of $SF_6$ have undergone a rapid increase, more than doubling from roughly 4.5 ppt in 2000 to 10 ppt in 2020 (Lan et al., 2024). In 2020, the radiative forcing of $SF_6$ was 5.9 mW/m$^2$ (Laube et al., 2023). This value could surge tenfold by the end of the 21st century if the upward trend in global $SF_6$ emissions persists, as pointed out by Hu et al. (2023).

$SF_6$ plays a crucial role in various industrial applications due to its remarkable insulating properties and chemical stability (e.g. Cui et al., 2024). It is primarily used in high-voltage electrical equipment in the power industry, such as gas-insulated switch gears (IEEE, 2012), transmission lines (Koch, 2008), and transformers (Gouda et al., 2012), where it acts as a dielectric and insulator. Here, emissions occur primarily during leakage, maintenance, and decommissioning of equipment (Zhou et al., 2018). Furthermore, $SF_6$ finds applications in semiconductor manufacturing, facilitating precise etching processes (Lee et al., 2004) and serves for blanketing or degassing in the magnesium or aluminum metal industry (Maiss and Brenninkmeijer, 1998). Moreover, it is used in medicine (Lee et al., 2017; Brinton and Wilkinson, 2009), photovoltaic manufacturing (Andersen et al., 2014), military applications (Koch, 2004), particle accelerators (Lichter et al., 2023), as a tracer gas (Martin et al., 2011), soundproof glazing (Schwarz, 2005), sports shoes (Pedersen, 2000), car tyres (Schwaab, 2000), wind turbines (EPA, 2023), and $SF_6$ measurements were used to determine OH radical concentrations in the stratosphere and troposphere (Li et al., 2018a).

$SF_6$ is regulated under the Kyoto Protocol. Thus, countries classified as ("developed") Annex-I nations must submit reports detailing their $SF_6$ emissions to the United Nations Framework Convention on Climate Change (UNFCCC). These national inventories are almost exclusively created by bottom-up methods, wherein statistical data of industrial production and consumption are used along with source-specific emission factors to estimate the emissions. However, $SF_6$ emissions have been shwn to be strongly underestimated by the bottom-up reports, underlining the need for independent verification methods (Levin et al., 2010). Therefore, bottom-up approaches such as inverse modeling on the basis of atmospheric measurements have been used in several studies to estimate $SF_6$ emissions (e.g., Brunner et al., 2017; Fang et al., 2014; Ganesan et al., 2014; Hu et al., 2023; Rigby et al., 2011; Simmonds et al., 2020; Vojta et al., 2022).

Around the year 2000, there was a notable shift in the global $SF_6$ emission pattern from a declining to an increasing trend, which has continued since then (Simmonds et al., 2020). This rising trend was primarily attributed to the increasing emissions



from ("developing") non-Annex-I Asian countries (Rigby et al., 2010). An inversion study by Fang et al. (2014) confirmed a
strong increase in East Asian SF$_6$ emissions between 2006 and 2009, and found its contribution to the global total emissions
to be 45%-49% between 2009 and 2012, with China being the largest contributor. Several other inversion studies identified
China as the major contributor to global SF$_6$ emissions (e.g., Ganesan et al., 2014; Rigby et al., 2011; Vojta et al., 2022). From
2007 to 2018 China's annual emissions increased from 1.4 to 3.2 Gg/yr accounting for 36% of the global total emissions in
2018, according to Simmonds et al. (2020). A recent inversion study by An et al. (2024) had access to data from a relatively
dense monitoring network inside China and estimated even higher Chinese emissions, with an increase from 2.6 Gg/yr in
2011 to 5.1 Gg/yr in 2021. Simmonds et al. (2020) also constrained Western European SF$_6$ emissions for the years 2013-2018
using three different regional inversion systems. Two of these inversion systems closely matched the emissions reported to the
UNFCCC, while the third one indicated substantially higher emissions. Brunner et al. (2017) found that Western European
SF$_6$ emissions were 47% higher than reported to the UNFCCC for the year 2011. As part of the UK annual report to the
UNFCCC, Manning et al. (2022) reported inversion results for SF$_6$ emissions in North-West Europe and found a decreasing
trend, dropping from 0.37 Gg/yr in 2004 to 0.18 Gg/yr in 2021. An atmospheric inversion study by Hu et al. (2023) found that
annual U.S. SF$_6$ emissions decreased between 2007 and 2018 but were on an annual basis 40 – 250% higher than calculated
by the U.S. Environmental Protection Agency's national inventory submitted to UNFCCC. They also suggested that U.S. SF$_6$
emissions were substantially higher in the winter than in the summer.

Up to this point, SF$_6$ inversion studies have exclusively been focusing on specific geographical areas, i.e., using regional
inversions only. Although global observation-based box models, such as the AGAGE 12-box model (e.g., Rigby et al., 2013)
are considered to be capable of accurately determining the global total emissions, a comprehensive top-down perspective of the
global SF$_6$ emission distribution is missing. Moreover, existing inversion studies often only use data from continuous surface
station measurements or from specific observation networks, potentially missing valuable information from other available
observations. In the absence of accurate global SF$_6$ mole fraction fields, many studies use statistical observation-based methods
to determine initial conditions for their inversions, which are suspected of introducing systematic errors in the inversion results
(Vojta et al., 2022). Lastly, the seasonality of SF$_6$ emissions has not been considered by inversion studies so far, with the
exception of the recent study by Hu et al. (2023).

Our study offers a comprehensive global, regionally resolved top-down perspective of SF$_6$ emissions, using inverse modeling
to determine the global emission distribution in the period between 2005 and 2021. We use all available SF$_6$ observations that
we could track down by merging continuous surface station measurements, flask measurements, and observations from aircraft
and ship campaigns. We consider multiple *a priori* emission fields for our inversion. For the initial conditions (Vojta et al.,
2022), we assimilate global SF$_6$ observations into modeled global three-dimensional SF$_6$ concentration fields, resulting in an
atmospheric SF$_6$ re-analysis for the period 2005-2021. We investigate regional and national SF$_6$ emission trends with annual
and also monthly resolution, and compare our results to various existing regional studies. Finally, we discuss our global total
emission trend and compare it to results from the AGAGE 12-box model and to global emissions directly calculated from
annual increases in globally-averaged atmospheric SF$_6$ mole fractions provided by NOAA (Lan et al., 2024).





## 2 Methods

### 2.1 Measurement data

The SF$_6$ re-analysis (Sec. 2.3) and the atmospheric inversion (Sec. 2.5) are based on globally distributed atmospheric observations of SF$_6$ dry-air mole fractions collected during the period 2005 to 2021. Our data set combines both continuous on-line and instantaneous flask sample measurements from surface stations, with observations from moving platforms. Figure 1 shows all surface station sites included in the inversion and the re-analysis. Figure A1 gives an overview of all the measurements from moving platforms, highlighting the measurement date and altitude with different colors. In addition, Section S3 as well

as Tables S1 (continuous surface stations), S2, S3 (flask measurement stations), and S4 (moving platforms) list all the data sets used and give further details. The measurements were provided by several independent organizations, and by international observation networks such as AGAGE and NOAA. Table S5 lists all the individual providers and their acronyms. Most of the data can be found in databases like WDCGG (di Sarra et al., 2022), EBAS (Tørseth et al., 2012), and CEDA (CEDA, 2023). We standardize all observations to the SIO-2005 calibration scale, as described in section S4.

For the inversion, continuous surface measurements were averaged over 3-hour intervals. Observations from moving platforms were averaged on a spatio-temporal grid with a temporal resolution of 3 hours and a spatial resolution of 0.5° in latitude, 0.5° in longitude, and 300 m in height. No observation averaging was performed for the re-analysis. Our complete dataset consists of around 2.7 million observations, while the averaged dataset comprises roughly 800,000 observations. Figure S1 shows the total number of annual observations available for (a) the entire dataset and (b) the averaged dataset.

### 2.2 Atmospheric transport

We use the Lagrangian particle dispersion model (LPDM) FLEXPART 10.4 (Pisso et al., 2019) to simulate the atmospheric transport of SF$_6$ between the emission sources and the measurement locations. The model does not account for removal processes, as SF$_6$ is almost inert in the troposphere to middle stratosphere. We run FLEXPART in backward mode releasing 50,000 particles continuously over 3-hour intervals from the measurement locations and tracking them backward in time for 50

days. For the continuous and moving platform observations, the 3-hour intervals are identical to the 3-hour averaging windows mentioned above (Sec. 2.1). For the flask measurements, the 3-hour intervals are centered around the measurement time. FLEXPART determines emission sensitivities shown as linear operator $\mathbf{H_e}$, which allows us to relate mole fraction values at the measurement location $\mathbf{y}$ with the corresponding emissions $\mathbf{e}$ occurring during the 50-day simulation period. The emissions prior to the simulation cannot be directly related, but still contribute to the measured mole fraction value and thus must be

accounted for in the model as well (Sec. 2.2.2). Therefore, FLEXPART also determines sensitivities to the initial conditions, which are shown as linear operator $\mathbf{H_i}$, which is multiplied by a 3-d SF$_6$ mole fraction field $\mathbf{y_i}$ (Sec. 2.3) 50 days before the respective measurement to obtain the baseline $\mathbf{H_i y_i}$. The relationship between receptor mole fractions $\mathbf{y}$, initial conditions $\mathbf{y_i}$ and emissions $\mathbf{e}$ is given by:



**Figure 1.** Locations of stations with continuous surface measurements (red triangles) and surface flask measurements (black dots) used in the inversion.

$$\mathbf{y} = \mathbf{H_e e} + \mathbf{H_i y_i} = \mathbf{Hx}, \tag{1}$$

where $\mathbf{H}$ is the complete atmospheric transport operator combining $\mathbf{H_e}$ and $\mathbf{H_i}$, and $\mathbf{x}$ is the state vector combining $\mathbf{e}$ and $\mathbf{y_i}$.

We run FLEXPART with hourly ECMWF ERA5 wind fields (Hersbach et al., 2018) with $0.5° \times 0.5°$ resolution, and 137 vertical levels. The global output grid has a resolution of $1° \times 1°$ and 18 vertical layers with interface heights at 0.1, 0.5, 1, 2, 3, 4, 5, 7, 9, 11, 13, 15, 17, 20, 25, 30, 40, and 50 km agl. The emission sensitivities were calculated only for the lowest layer

from 0 to 100 m agl, where most emissions occur.



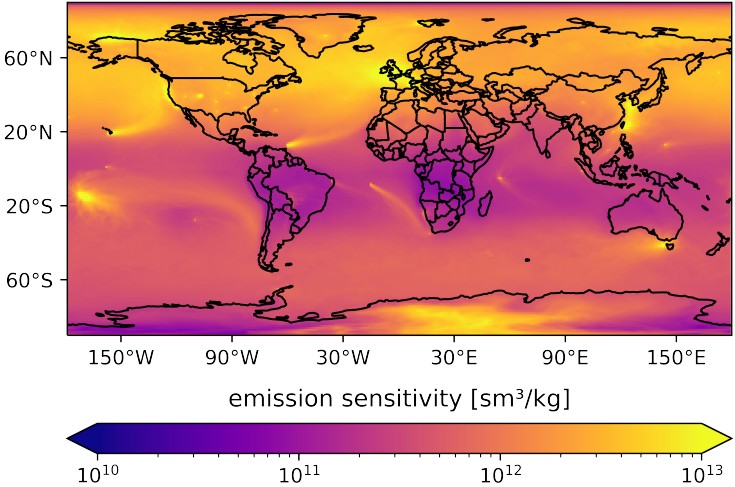

**Figure 2.** Annually averaged emission sensitivities for the example year 2019 obtained from FLEXPART 50-day backward simulations

### 2.2.1 Emission sensitivities

Figure 2 shows the annual averaged emission sensitivities for all observations made in the example year 2019. Areas of high sensitivity are well covered by the measurement data set, so that emissions can be well constrained by the inversion. Emission sensitivities in the Northern Hemisphere are much higher than in the Southern Hemisphere, and the high $SF_6$ emitting countries

China and USA are reasonably well covered. The largest values are observed in North-West Europe, which is very well monitored by the dense British observation network. However, large land areas in the Southern Hemisphere, including South America, Southern Africa, and Northern Australia, are poorly sampled due to a lack of continuous measurements. India, which is considered to have high $SF_6$ emissions, is also poorly covered. In these areas, the emissions cannot be determined well by the inversion.

### 2.2.2 Initial conditions

Using a LPDM to calculate emission sensitivities for atmospheric inversions, we release virtual particles directly from the measurement location and benefit from almost infinite resolution at the receptor. The disadvantage of using a LPDM is that we have to deal with initial conditions, as virtual particles can be followed backward only for a limited period, due to computational costs. Only emissions that occur within this LPDM simulation period can be directly related to observed mole fraction values

and are accessible to the inversion. We, therefore, need to define a baseline that accounts for all the emission contributions prior to the simulation period that contribute to the observed mole fraction. In this study, we use the global-distribution based (GDB) method (Vojta et al., 2022) to determine the baseline. We couple the mole fraction sensitivity at the ending points of the FLEXPART back trajectories to a global field of $SF_6$ mole fractions (for more details see Thompson and Stohl, 2014). In essence, this propagates the time-resolved 3-d mole fractions in space and time along the 50-day trajectories to the receptor



location and time. As pointed out by Vojta et al. (2022), the GDB method has many advantages over observation-based filtering methods. GDB baselines are consistent with the LPDM backward simulation length, account for meteorological variability, and allow the inclusion of low-frequency measurements and measurements from moving platforms in the inversion. However, the method requires unbiased global time-resolved 3-d fields of $SF_6$.

### 2.3   Global $SF_6$ fields

In this study, we generate global fields of $SF_6$ mole fractions for the period between 2005 and 2021, using the LPDM FLEX-PART 8-CTM-1.1 (Henne et al., 2018). The model is described by Groot Zwaaftink et al. (2018), who tested its performance for $CH_4$, while Vojta et al. (2022) applied it to $SF_6$. We operate FLEXPART-CTM in a domain-filling mode, where 80 million virtual particles are dispersed globally in proportion to air density. The initialization is based on a latitudinal $SF_6$ profile determined by interpolation of surface measurements and accounts for the "Age of Air" (Stiller et al., 2021) at higher altitudes

(for more details see Sec. S5 in the supplementary materials). Released particles are tracked forward in time and carry both an air tracer and the chemical species $SF_6$. When they reside in the atmospheric boundary layer, the model accounts for $SF_6$ emissions by increasing the $SF_6$ masses of the respective particles. The emission uptake of the particles is driven by the "UP" *a priori* emission data set (see Sec. 2.4).

     As model errors and inaccurate emission fields lead to errors and biases in the global $SF_6$ fields, a nudging routine is used to

push the simulated mole fractions towards the observations within predefined kernels centered around the measurement locations. We include the entire observation data set in the nudging routine, comprising continuous surface station measurements, flask measurements, and observations from aircraft and ship campaigns. Furthermore, we assign different kernel sizes to individual observations, according to the observed variability in a selected time window for stationary sites, and according to the measurement height for moving platforms. Small kernels are attributed to observations with higher variability and observations

close to the surface to preserve the spatial variability of $SF_6$ mole fractions over land masses. Detailed kernel configurations can be found in Table S6. We run the model with the 0.5°×0.5° ERA5 data set and produce daily average output with a resolution of 3°×2°. The daily-resolved global $SF_6$ mole fraction fields between 2005 and 2021 can be freely downloaded from https://doi.org/10.25365/phaidra.489.

### 2.4   *A priori* emissions

We generate six different annually resolved global $SF_6$ emission fields for the period 2005 to 2021 that are used as *a priori* emissions in the inversions (Sec. 2.5). One of these fields is also used to drive FLEXPART-CTM (Sec. 2.3). Our six *a priori* emissions are based on three different inventories (see Table 1) and globally gridded based on different proxy information at a resolution of 1°×1°.



**Table 1.** Overview of global SF$_6$ *a priori* emission fields used in this study

| *a priori* emissions | | |
|---|---|---|
| **Inventory** | **variation** | **distribution of total national emissions** |
| UNFCCC-ELE | UP | emissions distributed according to population density |
| | UN | emissions distributed according to night light remote sensing |
| EDGAR | E8 | v8 - distribution provided by EDGAR |
| | E7P | v7 - emissions distributed according to population density |
| | E7N | v7 - emissions distributed according to night light remote sensing |
| GAINS | GS | distribution provided by GAINS |

**UNFCCC-ELE**

For every year, we gather total national SF$_6$ emissions reported to the UNFCCC (UNFCCC, 2021) and add total Chinese emissions estimated by Fang et al. (2014). We then subtract the total emissions of these countries from the total global SF$_6$ emissions calculated by Simmonds et al. (2020). The residual emissions are then distributed among all other countries proportionally to their national electricity generation. Gaps in the SF$_6$ emissions or electricity generation data are filled by linear interpolation. Lastly, the attributed total national SF$_6$ emissions are further distributed within the respective borders of each country according to two different proxy data sets: (1) the gridded population density (CIESIN, 2018) (UP) and (2) night light remote sensing data (Elvidge et al., 2021) (UN), thus resulting in two different UNFCCC-ELE *a priori* emission versions.

**EDGAR**

We use the gridded annual global SF$_6$ emission inventory provided by the Emissions Database for Global Atmospheric Research (EDGAR, 2023; Crippa et al., 2023), part of the recently updated data set EDGARv8.0 (E8). In addition, we also utilize the national annual totals of SF$_6$ emissions provided by EDGARv7.0 (EDGAR, 2022; Crippa et al., 2021), which are not gridded. As for the UNFCCC-ELE emissions, we distribute those national totals according to the gridded population density (CIESIN, 2018) (E7P) or night light remote sensing (Elvidge et al., 2021) (E7N).

**GAINS**

Furthermore, we use the GAINS gridded global emission inventory. This inventory is based on the study by Purohit and Höglund-Isaksson (2017) and was updated until 2020 as described in Section S6 in the supplementary material. The provided data set was extended to 2021 by linear extrapolation (GS).

**Comparison**

Emission fields from the three inventories (UNFCCC-ELE, EDGAR, and GAINS) show much stronger differences than the two variations of UNFCCC-ELE and EDGAR generated by using different proxy information for spatial distribution. In Fig. 3,





we therefore only compare three *a priori* emissions (UP, E8, and GS) for 2019, as an example. It is noteworthy that these three emission fields show similar global total SF$_6$ emissions in 2019. Figure 3 shows significantly higher emissions in the Northern than in the Southern Hemisphere for all three fields, with China being the biggest emitter. Other high-emitting areas are Europe, the USA, and India. While emissions in Europe are comparable across all data sets, notable differences can be seen in other regions: (1) UNFCCC-ELE (electricity generation distributed data for non-reporting countries) shows relatively high emissions in India and the Southern Hemisphere compared to EDGAR and GAINS. (2) EDGAR shows higher emissions in the USA than the other two *a priori* fields, and (3) GAINS exhibits higher emissions in China than UNFCCC-ELE and EDGAR.

## 2.5 Inversion method

We employ the inversion framework FLEXINVERT+ (Thompson and Stohl, 2014) to calculate optimized emissions (*a posteriori* emissions). FLEXINVERT+ uses equation 1, the atmospheric transport operator $\mathbf{H}$, *a priori* emissions $\mathbf{x}_p$, initial conditions $\mathbf{y}_i$, and observed mole fractions $\mathbf{y}$ to minimize the cost function $\mathbf{J}$ (Eq. 2), which represents the negative exponent of the *a posteriori* emissions probability distribution, derived by Bayes' theorem (e.g., Tarantola, 2005). The *a posteriori* emissions defined by the maximum of the distribution are found by minimizing the mismatch between modeled and observed mole fractions weighted by the observation error covariance matrix $\mathbf{R}$, and the difference between emissions $\mathbf{x}$ and their *a priori* values $\mathbf{x}_p$ weighted by the *a priori* emission error covariance matrix $\mathbf{B}$:

$$\mathbf{J}(\mathbf{x}) = \frac{1}{2}(\mathbf{x} - \mathbf{x}_p)^T \mathbf{B}^{-1}(\mathbf{x} - \mathbf{x}_p) + \frac{1}{2}(\mathbf{Hx} - \mathbf{y})^T \mathbf{R}^{-1}(\mathbf{Hx} - \mathbf{y}), \tag{2}$$

We optimize emissions on a 6-monthly basis and average the results for each year to obtain annual emissions between 2005 and 2021. In addition to the emissions, we also optimize the baseline ($\mathbf{H_i y_i}$) in the inversion on a monthly basis. The uncertainty of the baseline is set to 0.15 ppt. The *a priori* emission uncertainty is estimated to be 70% of the *a priori* value in each grid cell with a minimum value of $1 \cdot 10^{-13} \frac{kg}{m^2 h}$. Correlations between emission uncertainties are accounted for using an exponential decay model with a spatial scale length of 250 km and a temporal scale length of 90 days. For the inversion, we use emission grids with different cell sizes (Fig. 4, Fig. S2, Fig. S3), defined by the aggregation of grid cells with low emission contributions based on emission sensitivities and *a priori* emissions (see Thompson and Stohl, 2014, for a detailed description). We also exclude grid cells over the oceans from the inversion. The global inversion grid has a resolution of 1° to 16°, and the total number of grid cells varies between years, ranging from a minimum of 5841 (2005) to a maximum of 11901 (2016). To study the seasonal emission patterns, we also perform monthly inversions, using a coarser global inversion grid of 953 grid cells for all years and a time scale length of 30 days for the correlation between *a priori* emission uncertainties.

For SF$_6$ we only expect positive fluxes over land. However, the inversion algorithm may create negative *a posteriori* fluxes. To address this issue, we apply an inequality constraint on the *a posteriori* emissions, using the truncated Gaussian approach by Thacker (2007). *A posteriori* emissions $\hat{\mathbf{x}}$ are corrected to positive values by applying inequality constraints as error-free observations:

$$\hat{\mathbf{x}} = \mathbf{x} + \mathbf{AP}^T (\mathbf{PAP}^T)^{-1}(\mathbf{c} - \mathbf{Px}), \tag{3}$$



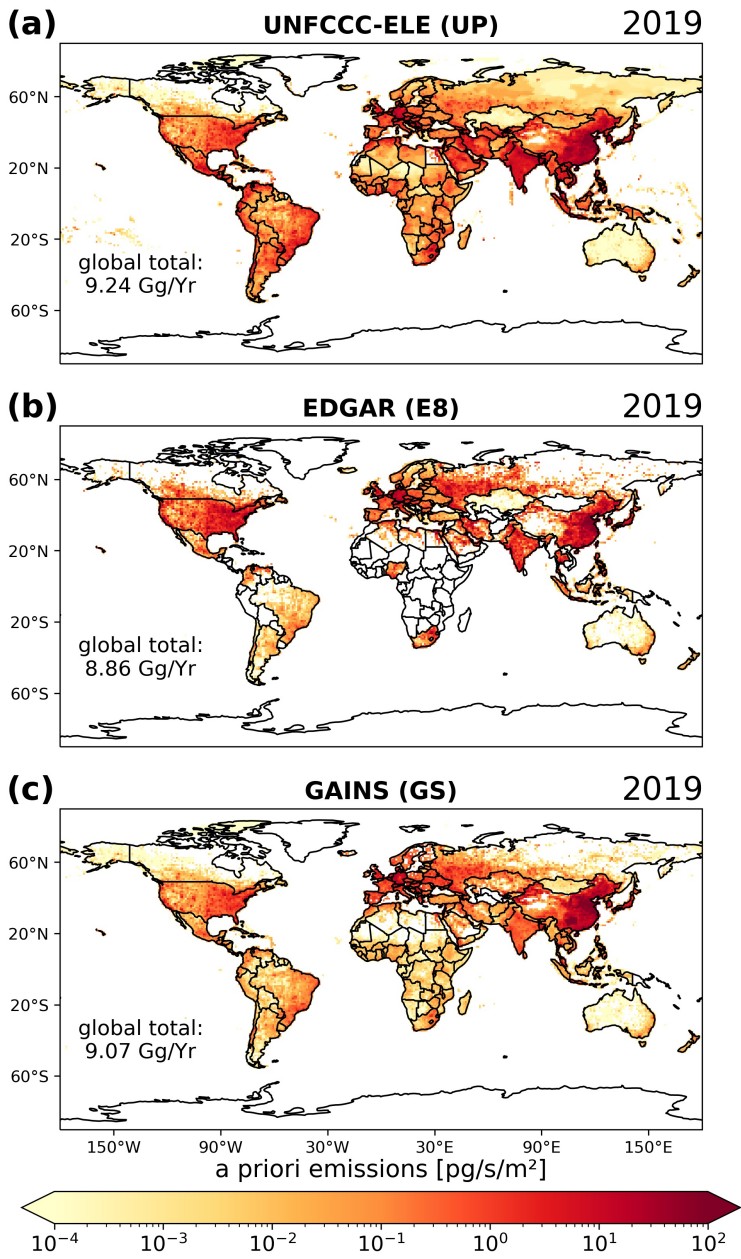

**Figure 3.** *A priori* emissions from the different sources (a) UNFCCC-ELE (UP), (b) EDGAR (E8), and (c) GAINS (GS) for the year 2019.

where $\mathbf{P}$ represents a matrix operator selecting the fluxes violating the inequality constraint, and $\mathbf{c}$ a vector of the inequality constraint. $\mathbf{x}$ and $\mathbf{A}$ represent the *a posteriori* emissions and error covariance matrix, respectively.



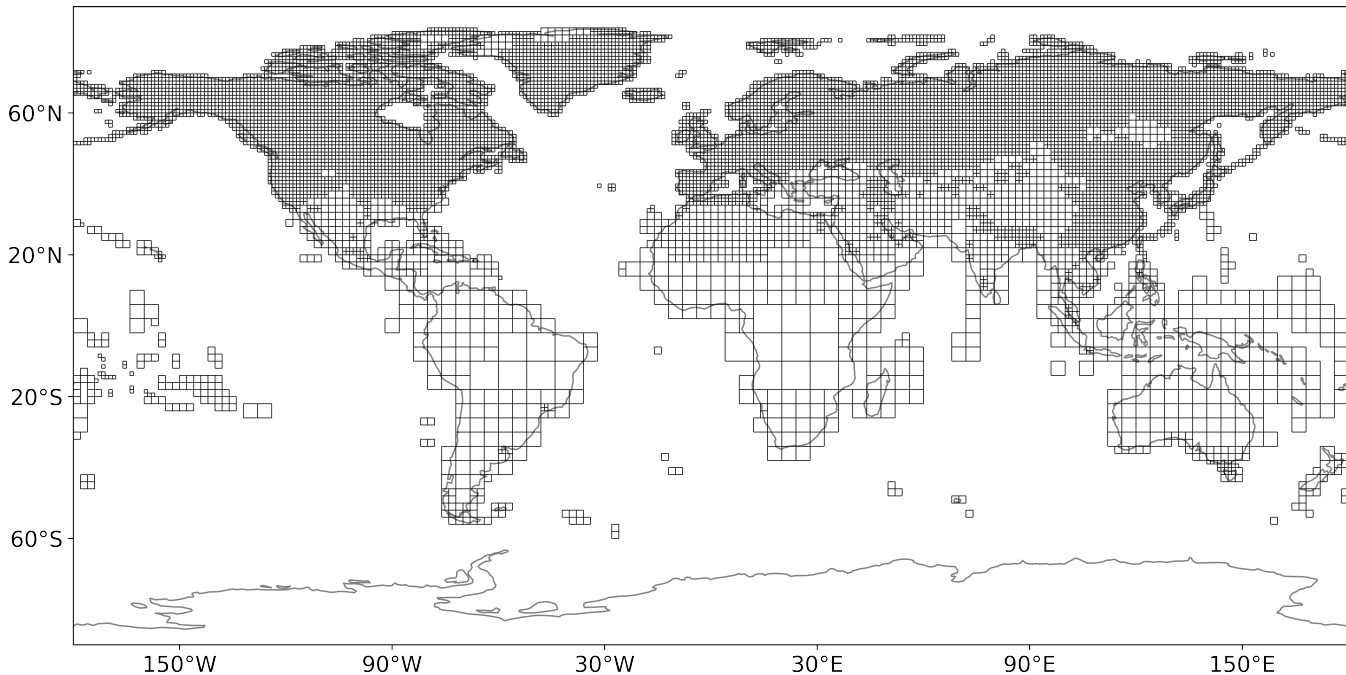

**Figure 4.** Global inversion grid with variable grid cell sizes for the example year 2019.

## 2.6 Sensitivity tests and setup

Before deciding on our final inversion setup, we performed several sensitivity tests. We tested different: (1) *a priori* emission
uncertainties between 50% and 100% of the respective *a priori* values and minimal absolute uncertainties between $1 \cdot 10^{-14}$
and $1 \cdot 10^{-12} \frac{kg}{m^2 h}$, (2) spatial and temporal correlation scale lengths of the *a priori* uncertainties of 100 to 300 km, and 30 to
180 days, respectively, and (3) baseline uncertainties from 0.05 to 0.25 ppt. We found that inversion results were relatively
stable for these different settings and that the choice of the *a priori* emission inventory (UNFCCC-ELE, EDGAR, or GAINS)

showed the biggest influence on the inversion results. While the inversion results were similar using different variations of the
UNFCCC-ELE (UP and UN) or EDGAR (E8, E7P, and E7N) *a priori* emissions (see Sec. 2.4), we found substantial differences
when switching between UNFCCC-ELE, EDGAR and GAINS. Therefore, we ran inversions with all six variations listed in
Table 1 individually and averaged the results of UP and UN, as well as E8, E7P, and E7N to compile one inversion result for
each *a priori* emission inventory (UNFCCC-ELE, EDGAR, and GAINS). Since it is challenging to identify the most accurate

inventory, we also provide an average of these three inversion results.



## 3 Results and discussion

### 3.1 Observed and modeled mole fractions

To illustrate the inversion optimization process, we compare observed and modeled mole fractions at the Gosan observation station (Fig. 5a), at the Ragged Point station (Fig. 5b), and all other continuous surface measurement sites (Fig. A2, A3, and

S4-S23), using the E7P emissions field. The Gosan station is situated on the southwestern tip of the South Korean island Jeju, monitoring pollution events from East Asia. However, during the Asian summer monsoon, typically from June to September, clean air from the Southern Hemisphere, low in $SF_6$, is episodically passing over the station (e.g. Li et al., 2018b), making it challenging to accurately define the baseline during this period. The background station Ragged Point, located on Barbados' eastern edge, primarily receives clean air masses from the Atlantic. It also exhibits intrusions of southern air masses that are low

in $SF_6$ during the summer, resulting in distinct minima in the mole fraction time series, and a complex baseline. With the GDB method, we can address these challenges of complex baselines. As illustrated in Fig. 5, the calculated baselines capture the low summer observations, representing a significant advantage over statistical baseline methods. This advantage also becomes apparent for other stations with complex baselines such as Hateruma (Japan, Fig. A2) or Izaña (Tenerife, Fig. A3). Additionally, the optimization of the baseline shows relatively little impact at all stations, implying that the GDB method and the utilized

global $SF_6$ mole fraction fields already lead to a well-fitting baseline that cannot be improved substantially by the inversion. This is important, as the optimization can focus on improving the emissions rather than correcting a wrong baseline. Figure 5a also illustrates the emission improvement achieved by the inversion. The optimized *a posteriori* emissions result in mole fractions that are much closer to the observations than the *a priori* modeled values. For Gosan, the correlation ($r^2$) between (detrended) observed and modeled values improves from 65% to 81% and the mean squared error (MSE) halves from 0.4 ppt$^2$

to 0.2 ppt$^2$. Table S7 and Fig. S24 demonstrate the statistical improvements at all continuous surface stations, emphasizing the proper functioning of the inversion. Figure 5b further illustrates the advantage of choosing a rather long 50-day backward simulation period. With this long simulation period, we can see that this remote station is also directly influenced by emissions (i.e., enhancements over the baseline) that can be directly optimized. With shorter simulation times (e.g., 5-10 days), no emission contributions above the baseline could be seen, thus rendering this station useless for emission optimization. For a

detailed discussion about the LPDM backward simulation period see Vojta et al. (2022).

### 3.2 Inversion increments and relative error reduction

Figure 6 shows the inversion increments (*a posteriori* minus *a priori* emissions) and the relative uncertainty reductions ($1 - \frac{a\ posteriori\ uncertainty}{a\ priori\ uncertainty}$) achieved by the inversion for the example year 2019, when using the *a priori* emission fields UP (UNFCCC-ELE), E8 (EDGAR) and GS (GAINS). Across all cases, the emission optimization predominantly occurs in

the Northern Hemisphere, characterized by non-zero inversion increments and large error reductions. The limited number of observations in the Southern Hemisphere results in small emission sensitivities there (see Fig. 2), limiting the effects of the inversion primarily to Northern Hemisphere emissions. Only in the case of the UNFCCC-ELE inventory, Fig. 6a shows (negative) inversion increments and notable error reduction in Southern regions like South America and South Africa. This might





**Figure 5.** Mole fraction time series at the (a) Gosan and (b) Ragged Point measurement station. Red lines represent the modeled *a priori* mole fractions calculated with the E7P *a priori* emissions and blue lines represent the modeled *a posteriori* mole fractions. The green line illustrates the baseline derived by the GDB method and the orange line shows the optimized baseline. The grey line represents the observed mole fractions. The inset panels zoom into the year (a) 2019 (Gosan) and (b) 2020 (Ragged Point), as illustrated by the light green rectangles.

indicate that the UNFCCC-ELE *a priori* emissions are significantly overestimated in these areas. All three data sets show the biggest error reduction and inversion increments in the USA, Europe, and China, where the *a priori* emissions are high and many observations are available. While the increments look similar for the three *a priori* emissions for Europe and China, they are very different for the USA, where the inversion produces predominantly negative increments when using the EDGAR inventory, while only positive increments are obtained using UNFCCC-ELE and GAINS. These differences suggest that the true 2019 U.S. emissions lie between the high EDGAR and the lower UNFCCC-ELE/GAINS estimates.





**Figure 6.** Inversion increments (*a posteriori* minus *a priori* emissions; left panels) and the relative uncertainty reductions (right panels) shown when using the priors (a) UNFCCC-ELE (UP), (b) EDGAR (E8) and (c) GAINS (GS), for the example year of 2019.

**3.3  National and regional emissions**

Figure 7 illustrates the global $SF_6$ *a posteriori* emissions for the example year 2019, averaged over all emission fields as described in Sec. 2.6. The highest $SF_6$ emissions can be seen in the USA, Europe, China, and India, while emissions are



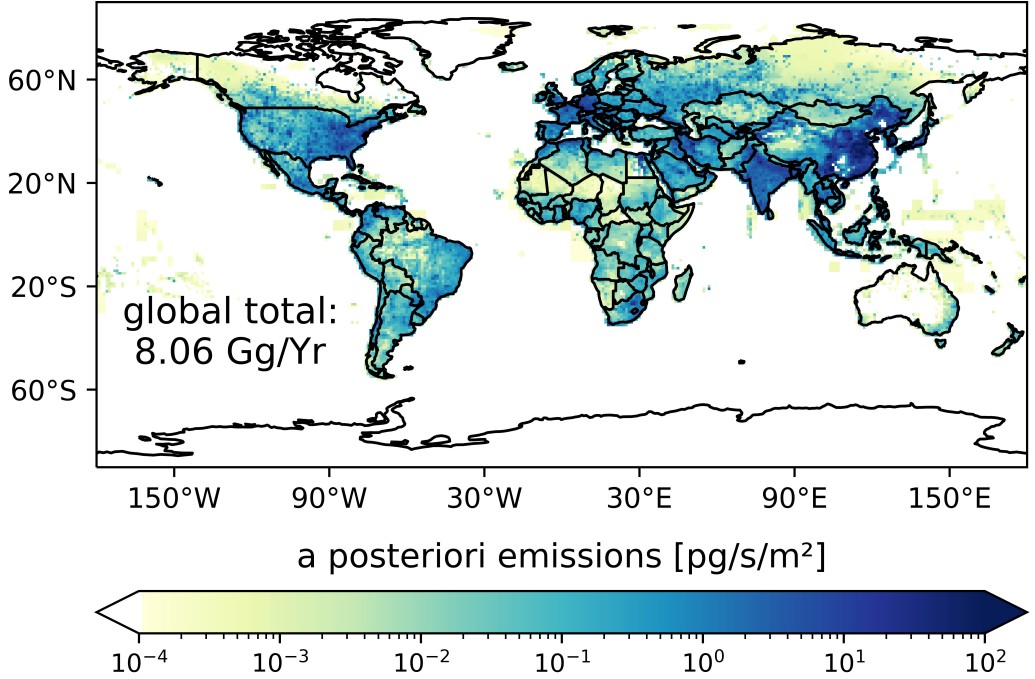

**Figure 7.** Global *a posteriori* emissions for the example year 2019, averaged over the inversion results using the six different *a priori* emissions.

smaller in South America, Africa, and Australia. SF$_6$ emissions of these countries and regions are discussed in more detail in the following subsections, showing their national/regional emission time series between 2005 and 2021. National and regional emissions are calculated by aggregating the emissions within the respective grid cells of the corresponding country or region, employing a national identifier grid (CIESIN, 2018).

### 3.3.1 Emissions from the United States of America

Figure 8 shows the annual *a priori* and *a posteriori* U.S. SF$_6$ emissions for the different priors in the period between 2005 and 2021. The inversion results show a clearly declining annual emission trend of -0.054 Gg/yr, dropping from 1.25 Gg in 2005 to 0.48 Gg in 2021 (Fig. 8; *a posteriori* average). However, the *a posteriori* emissions are larger (by a factor of 2 on average) than the emissions reported to UNFCCC (Fig. 8; *a priori* UNFCCC-ELE) throughout the entire study period. While the different *a priori* emissions show big differences, *a posteriori* emissions agree within their 1-$\sigma$ uncertainties. At the beginning of the study period, all three *a posteriori* emissions are substantially higher than the UNFCCC-reported *a priori* emissions, and closer to the EDGAR *a priori* estimates. Between 2005 and 2012 the *a posteriori* emissions show a substantial decrease, after which they approach the UNFCCC-reported values, but still remain higher. It also seems that the GAINS *a priori* emissions are far too low at the beginning of our study period, while the EDGAR *a priori* emissions are far too high at the end of our study

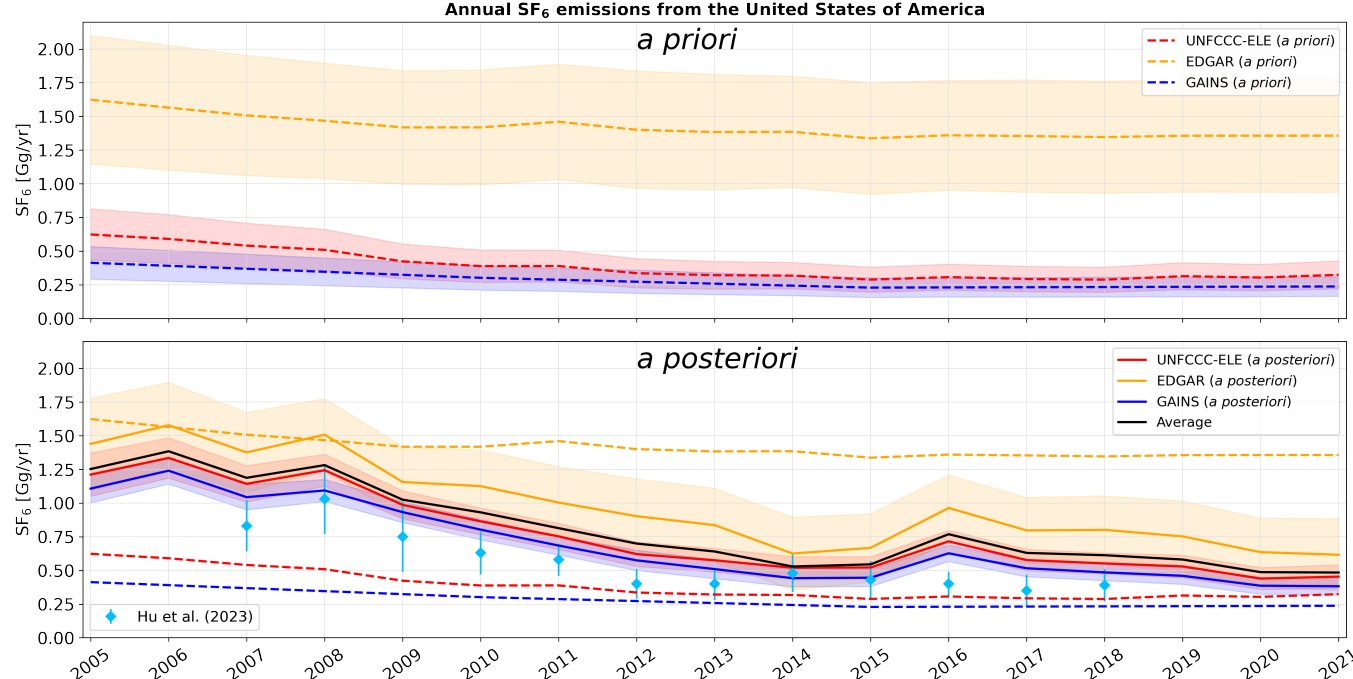

**Figure 8.** Annual *a priori* (dashed lines) and *a posteriori* (solid lines) SF$_6$ emissions in the U.S. for the period between 2005 and 2021 when using different *a priori* emissions (UNFCCC-ELE red, EDGAR orange, GAINS blue). The *a priori* emissions (top panel) and *a posteriori* emissions (bottom panel) are shown together with their respective 1-$\sigma$ uncertainties (colored shadings). The bottom panel also shows the average *a posteriori* emissions (black solid line) and the results of Hu et al. (2023), which are shown with blue diamonds and vertical lines representing their 2-$\sigma$ uncertainties. For better comparison, the *a priori* emissions (without uncertainties) are also included in the bottom panel.

period. Our results are a bit higher compared to the regional inversion study by Hu et al. (2023), however, show a remarkably similar declining trend in U.S. SF$_6$ emissions between 2007 and 2018. This good agreement with a regional inversion study focussing on the U.S. with a very different setup is reassuring.

### 3.3.2 Total emissions from EU countries

Figure 9 illustrates the total annual *a priori* and *a posteriori* SF$_6$ emissions from all EU countries[1]. Here, the three *a priori* data sets show almost no trend and are very similar to each other throughout the study period, indicating a consistent framework for bottom-up reporting of EU emissions. The annual *a posteriori* emissions show a decreasing trend of -0.006 Gg/yr, dropping from 0.41 Gg in 2005 to 0.25 Gg in 2021 (Fig. 9; *a posteriori* average). While *a posteriori* emissions are relatively stable and exceed the *a priori* emissions until 2017, there is a significant drop in 2018, after which they are closer to the *a priori*

---

[1]Austria, Belgium, Bulgaria, Croatia, Cyprus, Czech Republic, Denmark, Estonia, Finland, France, Germany, Greece, Hungary, Ireland, Italy, Latvia, Lithuania, Luxembourg, Malta, Netherlands, Poland, Portugal, Romania, Slovakia, Slovenia, Spain, and Sweden



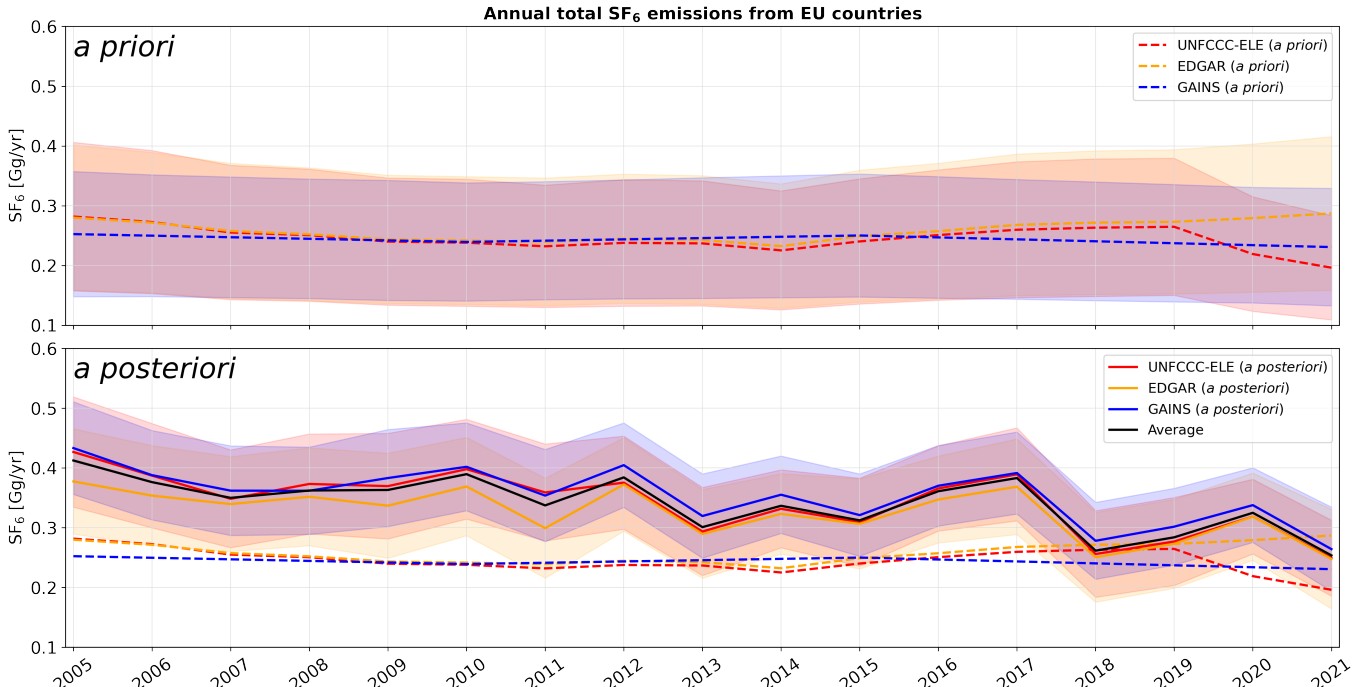

**Figure 9.** Annual *a priori* (dashed lines) and *a posteriori* (solid lines) $SF_6$ emissions aggregated for all EU countries, shown for the period between 2005 and 2021 when using different *a priori* emissions (UNFCCC-ELE red, EDGAR orange, GAINS blue). The *a priori* emissions (top panel) and *a posteriori* emissions (bottom panel) are shown together with their respective 1-$\sigma$ uncertainties (colored shadings). The bottom panel also shows the average *a posteriori* emissions with a black solid line. For better comparison, the *a priori* emissions (without uncertainties) are also included in the bottom panel.

emissions. It seems plausible that this drop in $SF_6$ emissions in 2018 was a result of the EU's F-gas regulation 517/2014 (European Parliament and Council of the European Union, 2014), which requires new electrical switch gear put into service from 2017 onwards to be equipped with a leak detection system and bans the use of $SF_6$ for recycling magnesium die-casting alloys from 2018. Our results suggest that in their reports to the UNFCCC, EU countries underestimated their $SF_6$ emissions

prior to 2018, but at the same time underestimated the positive effect of the F-gas regulation 517/2014 in cutting $SF_6$ emissions.

As one of only three countries, the United Kingdom also includes top-down inversion results in its annual UNFCCC reports (Manning et al., 2022). As part of this top-down approach, Manning et al. (2022) also reported emissions of North-West Europe[2], to which we compare our inversion results (Fig. A4). The *a posteriori* emissions from North-West Europe are generally similar to EU emissions shown in Fig. 9, however, they show an even clearer negative trend of -0.009 Gg/yr. Our results agree

well, on average within 16% and better since 2012, with those reported by Manning et al. (2022). Furthermore, Simmonds et al. (2020) presented inversion-derived emissions for Western Europe[3] for four different inversion setups. Our *a posteriori*

---

[2]Ireland, the United Kingdom of Great Britain, France, Belgium, Netherlands, Luxembourg, and Germany

[3]United Kingdom of Great Britain, Ireland, Benelux, Germany, France, Denmark, Switzerland, Austria, Spain, Italy, and Portugal



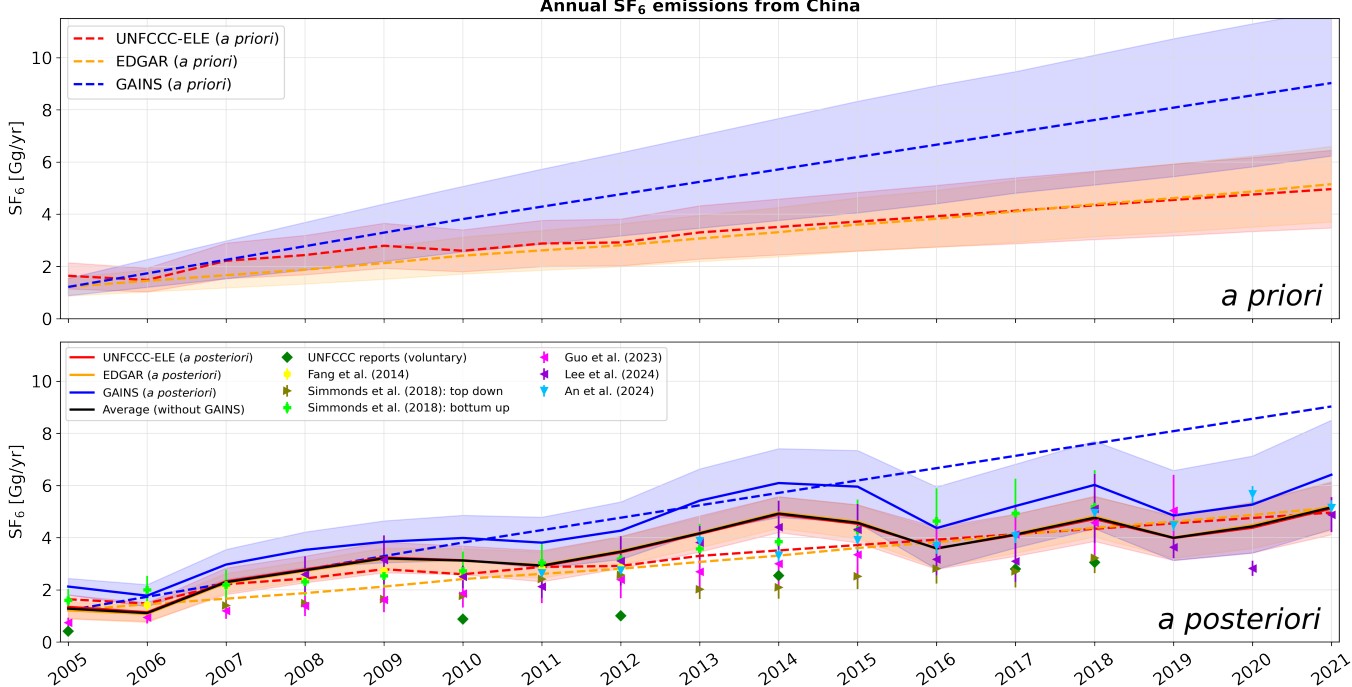

**Figure 10.** Annual *a priori* (dashed lines) and *a posteriori* (solid lines) $SF_6$ emissions from China in the period between 2005 and 2021 when using different *a priori* emissions (UNFCCC-ELE red, EDGAR orange, GAINS blue). The *a priori* emissions (top panel) and *a posteriori* emissions (bottom panel) are shown together with their respective 1-$\sigma$ uncertainties (colored shadings). The bottom panel also shows the average *a posteriori* emissions (excluding GAINS) with a black solid line, together with various reference values. For better comparison, the *a priori* emissions (without uncertainties) are also included in the bottom panel.

emissions agree very well with three of these four inversions (Fig. A5). The fourth inversion shows consistently lower emissions, however, this inversion setup used fewer observation stations than the other three and is likely less accurate. It is likewise noteworthy that the first three inversions of Simmonds et al. (2020) show an emission drop in 2018, which we also find.

### 3.3.3 Emissions from China

Chinese *a priori* and *a posteriori* $SF_6$ emissions are illustrated in Fig. 10. The inversion-derived *a posteriori* emissions reveal a distinct positive trend of 0.21 Gg/yr (Fig. 10; *a posteriori* average without GAINS), with a particularly rapid increase between 2006 and 2014 (0.35 Gg/yr), followed by a stabilization thereafter. The UNFCCC-ELE *a priori* Chinese emissions slightly exceed the EDGAR *a priori* emissions between 2007 and 2011, after which they align well. UNFCCC-ELE and EDGAR *a posteriori* emissions show almost identical Chinese emissions that are also close to their *a priori* values. The GAINS *a priori* Chinese emissions differ significantly from the other two inventories. After 2005, the GAINS *a priori* emissions show a very strong upward trend, increasingly diverging from the other two priors until the end of the study period, at which point





the GAINS Chinese emissions are almost twice as high as the other priors. In the GAINS inventory, China's 2021 emissions alone would account for almost all of the known total global $SF_6$ emissions (see Sec. 3.3.5), which seems unrealistic. The
GAINS *a posteriori* emissions for China show lower values compared to the *a priori* emissions, however still exceed the UNFCCC-ELE- and EDGAR-derived results, even though there is an overlap in the uncertainty bands. It seems likely that the inversion improves the overestimated Chinese GAINS emissions, yet it may not entirely correct them, given the considerably overestimated *a priori* estimates. Due to these concerns about the Chinese GAINS *a priori* emissions, we provide both a Chinese *a posteriori* emissions average including (see Table A3) and excluding GAINS inversions (black solid line in Fig. 10).
China is not obliged to report its national emissions but it voluntarily reported bottom-up $SF_6$ estimates in their national communications and biennial updates to the UNFCCC for 2005 (China, 2012), 2010 (China, 2018a), 2012 (China, 2016), 2014 (China, 2018b), 2017 (China, 2023a), and 2018 (China, 2023b). These reported values are much smaller than our *a posteriori* emissions, especially in 2010, 2012, and 2014. We also compare our results to various other studies of Chinese emissions, both using bottom-up and top-down approaches. Our results agree within 15% with the inversion study by Fang et al. (2014)
who used a similar inversion setup, based on the continuous measurements in Gosan (South Korea), Hateruma (Japan) and Cape Ochiishi (Japan), and FLEXPART atmospheric transport modeling. Furthermore, our results align closely with a recent inversion study by An et al. (2024) (agreeing within 12%), who had access to data from a relatively dense monitoring network over China. Our results also agree well (within 15%) with the findings of (Lee et al., 2024), whose regional inversion study (in preparation) utilizes observations from Gosan to estimate emissions in South-East Asia. Note that the patterns of our time series
are very similar to the ones of Lee et al. (2024), suggesting that our Chinese *a posteriori* emissions are highly influenced by the Gosan observations station. Our derived emissions also agree well within 8% with bottom-up estimates by Guo et al. (2023) after 2015 and within 18% with the bottom-up estimates by Simmonds et al. (2020). Our results are, however, higher than the bottom-up estimates by Guo et al. (2023) between 2008 and 2015 and the inversion-derived emissions by Simmonds et al. (2020). However, Simmonds et al. (2020) based their inversion results on only one station (Gosan), coarser meteorology, and
an inversion domain representing only 34% of China's population, which could have resulted in a substantial underestimation of the emissions (An et al., 2023).

### 3.3.4 Other regions

In this section, we present the *a priori* and *a posteriori* $SF_6$ emissions from Africa, South America, Australia, and India. It is important to note that there are no emission reports to the UNFCCC for Africa, South America, and India. In these regions,
the UNFCCC-ELE *a priori* emissions are derived by distributing the emissions residuals from the global total emissions (Simmonds et al., 2020) when subtracting the cumulative reported emissions from Annex-I countries, according to the national electricity generation as described in Sec. 2.4.

**Africa**

Figure 11 shows African *a priori* and *a posteriori* $SF_6$ emissions. One can see that the GAINS inventory is very low and the
UNFCCC-ELE inventory is very high in comparison to the EDGAR inventory. Before 2018, the UNFCCC-ELE *a posteriori*



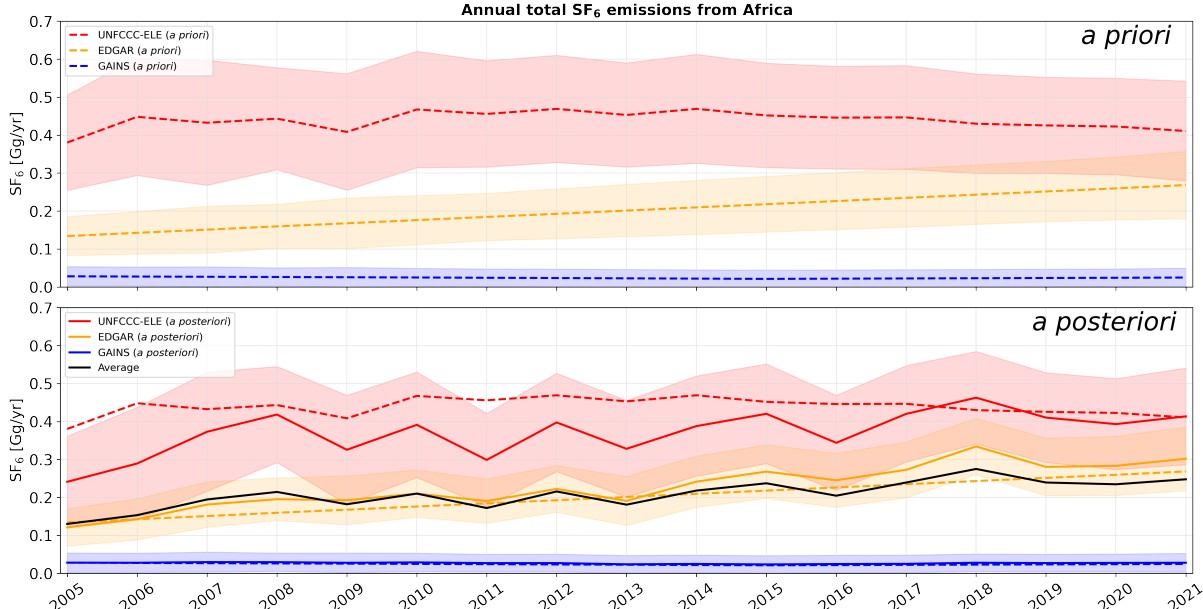

**Figure 11.** Annual *a priori* (dashed lines) and *a posteriori* (solid lines) $SF_6$ emissions from Africa, shown for the period between 2005 and 2021 when using different *a priori* emissions (UNFCCC-ELE red, EDGAR orange, GAINS blue). The *a priori* emissions (top panel) and *a posteriori* emissions (bottom panel) are shown together with their respective 1-$\sigma$ uncertainties (colored shadings). The bottom panel also shows the average *a posteriori* emissions with a black solid line. For better comparison, the *a priori* emissions (without uncertainties) are also included in the bottom panel.

emissions are lower than the *a priori* values and align with them afterwards. EDGAR *a posteriori* emissions are overall higher than the respective *a priori* emissions. It seems likely that the inversion improves the UNFCCC-ELE overestimation and the EDGAR underestimation, however cannot entirely correct them, as large parts of Africa are poorly covered by the observation network (see Fig. 2). The GAINS *a posteriori* emissions are consistently higher than the GAINS *a priori* emissions but the

increases are very small. It seems that the GAINS *a priori* emissions are too small and the inversion tries to increase them but is bound by the low uncertainties assumed, resulting only in minor corrections. Thus, even the GAINS *a posteriori* likely underestimate the true emissions. Note that both, UNFCCC-ELE and EDGAR *a posteriori* emissions show a larger positive trend than the *a priori* emissions. This is also true for the GAINS prior, however differences are very small. The averaged *a posteriori* emissions are close to the EDGAR inventory and show a slowly increasing trend of 0.006 Gg/yr, growing from

0.13 Gg in 2005 to 0.25 Gg in 2021.

**South America**

For South America (see Fig. A6), the UNFCCC-ELE inventory is more than 10 times higher than the EDGAR and GAINS inventory, and GAINS is on average 38% higher than EDGAR. Due to the narrow uncertainty bands and the poor observa-





tional coverage of South America, the inversion results stay close to the *a priori* emissions for EDGAR and GAINS. For

UNFCCC-ELE *a posteriori* emissions are smaller than the *a priori* values, especially at the beginning of the study period. We therefore suspect a substantial overestimation by the UNFCCC-ELE *a priori* inventory, given that the UNFCCC-ELE *a posteriori* emissions are partly lowered considerably, despite the poor coverage. Note also that UNFCCC-ELE inversion results show a positive trend of 0.007 Gg/yr, in contrast to the *a priori* inventory.

**Australia**

Figure A7 shows Australian *a priori* and *a posteriori* $SF_6$ emissions. All *a priori* emission inventories show similar values throughout the whole study period, well below 0.01 Gg/yr. The relatively wide uncertainty bands result from the chosen minimal *a priori* uncertainty, which is assigned to grid cells with low emissions (see Sec. 3), providing the algorithm with more freedom to deviate from the *a priori* emissions. Nevertheless, inversion results stay close to the *a priori* values. This is to be expected given that there are no $SF_6$ measurements available within the country, except the Cape Grim station in Tasmania,

which predominantly captures clean air from the Indian Ocean.

**India**

India can be identified as the most challenging region for $SF_6$ inverse modeling, where *a priori* emission inventories show substantial differences but where emissions could be of global significance (UNFCCC-ELE emissions are about 8% of global emissions in 2021) (Fig. A8). For the UNFCCC-ELE inventory, Indian inversion increments are much higher compared to

EDGAR or GAINS (see Fig. 6), resulting in large discrepancies across the *a posteriori* emissions of the different inventories (Fig. A8). This can be related to the poor observational coverage (see Fig. 2) in combination with the relatively high UNFCCC-ELE *a priori* uncertainties, which might allow the algorithm to excessively relate the distant high East Asian measurements to Indian emissions. The GAINS inventory shows by far the lowest Indian *a priori* emission, while inversion results stay very close to the prior values, due to the small *a priori* uncertainty bands. However, all inversions show a much stronger trend in *a*

*posteriori* $SF_6$ emissions than in the *a priori* emissions. A strong upward trend in $SF_6$ emissions may indeed be expected given that the installed electric power generation capacity in India has almost quadrupled between 2002 and 2022 (Government of India, 2023)

### 3.3.5 The global perspective

Our study aimed to incorporate all globally accessible $SF_6$ observations in the inversion, in combination with long backward

trajectories of 50 days to make the best use of the observation network (Vojta et al., 2022). These are optimal conditions for constraining both regional and global $SF_6$ emissions. To judge the quality of our *a posteriori* global emission, we compare our results with the global emissions calculated by Simmonds et al. (2020) for the years 2005 to 2018 using the AGAGE 12-box model (e.g., Rigby et al., 2013), which we linearly extrapolated until 2021. Such box models are considered to be capable of constraining the global total $SF_6$ emissions within a few percent, because the average atmospheric growth rate can be measured



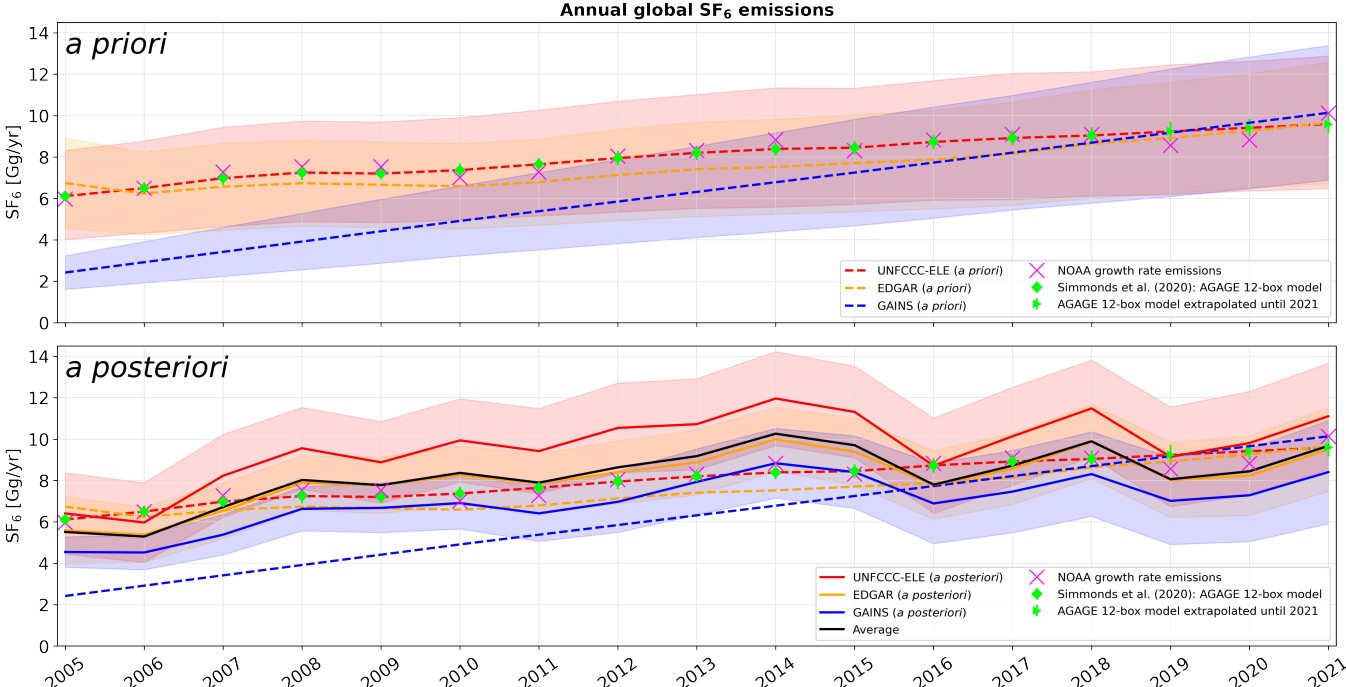

**Figure 12.** Annual total global *a priori* (dashed lines) and *a posteriori* (solid lines) SF$_6$ emissions in the period between 2005 and 2021. The *a priori* emissions (top panel) and *a posteriori* emissions (bottom panel) are shown together with their respective 1-$\sigma$ uncertainties (colored shadings). The bottom panel also shows the average *a posteriori* emissions with a black solid line. Reference values of the AGAGE 12-box model (linearly extrapolated until 2021) and NOAA growth rate emissions are shown with green diamonds/rectangles and purple crosses, respectively. For better comparison, the *a priori* emissions (without uncertainties) are also included in the bottom panel.

accurately and the very long atmospheric lifetime of SF$_6$ leads to small uncertainties in global total emissions. In addition, we compare our results with global emissions directly calculated from annual increases in globally-averaged atmospheric SF$_6$ mole fractions provided by NOAA (Lan et al., 2024), which we multiply by the factor $\frac{M_{SF_6}}{M_{air}} \cdot m_{atm}$, where $M_{SF_6}$ and $M_{air}$ represent the molecular weights of $SF_6$ and air, and $m_{atm}$ is the mass of the atmosphere. We refer to these emissions as "NOAA growth rate emissions".

Figure 12 illustrates the *a priori* and *a posteriori* total global SF$_6$ emissions, compared to the reference values of the AGAGE 12-box model and the NOAA growth rate emissions. In general, the NOAA growth rate emissions agree well with the box model, however, show more temporal variability. The UNFCCC-ELE *a priori* global emissions coincide per definition with the AGAGE 12-box model (Sec. 2.4), while the UNFCCC-ELE *a posteriori* global emissions are on average 16% higher. The uncertainties stated for the AGAGE 12-box model are only about 3%, with an additional 1% that may be attributed to SF$_6$

lifetime uncertainties (Simmonds et al., 2020), while our uncertainties are higher. The two estimates are within the combined uncertainties for most individual years but overall our UNFCCC-ELE *a posteriori* global emissions seem to be systematically



too high. One possibility for explaining this discrepancy is a potential ocean sink of $SF_6$ that is not accounted for in the AGAGE 12-box model, leading to a potential underestimation of global emissions in the box model. Ni et al. (2023) recently suggested that such an ocean sink may account for about 7% of the global $SF_6$ emissions. We tested this hypothesis by allowing for an

oceanic sink in our inversion. However, the inversion-derived oceanic *a posteriori* emissions showed either a lot of noise or no fluxes at all (in case of optimizing ocean fluxes in one aggregated oceanic grid cell). Therefore, we were unable to confirm the presence of oceanic $SF_6$ sinks with our inversion. Yet, another possible explanation for the increase of the global emissions by the inversion is the positivity constraint employed on the emissions over land, which might lead to a positive bias of the *a posteriori* global emissions. However, tests showed that the positivity constraint on the *a posteriori* emissions had very little

effect (<1%) on the total global emissions. There is a better explanation for our too-high *a posteriori* emissions. As discussed in Sec. 3.2 the measurement data puts relatively strong constraints on the high emitting regions China, Europe and USA that are responsible for the biggest part of the global $SF_6$ emissions. National inversion results showed that reported UNFCCC emissions in these regions are predominantly underestimated. Consequently, to match the global total emission, our UNFCCC-ELE inventory attributed too high emissions to countries not reporting their emissions to the UNFCCC (e.g., in South America,

Africa or India). Unfortunately, the emissions in these regions are very poorly constrained by the existing observation network (see Fig. 2). As shown in Sec. 3.3.4, the inversion can reduce large biases in these regions but we cannot expect it to remove them completely, and this leads to a positive bias in *a posteriori* global emissions.

The global GAINS *a priori* emissions are lower than all other inventories at the beginning of the study period, and its positive trend is larger and inconsistent with the global atmospheric $SF_6$ growth postulated by the box model and the NOAA

measurements. Due to this rapid increase, the GAINS *a priori* emissions converge with the other emission inventories by the end of the study period. The global GAINS *a posteriori* emissions are much closer to the AGAGE box model results and NOAA growth rate emissions than the *a priori* emissions and align well with their trends. However, *a posteriori* emissions are 15% lower on average, indicating that aggregated emissions are underestimated in poorly monitored areas. This claim can be supported by comparing the global GAINS and Chinese GAINS *a priori* emissions (Fig. 10). At the beginning of the

study period GAINS seems to produce realistic Chinese emissions, while at the same time, global emissions are significantly underestimated. After rapid growth, global emissions are close to the reference box model value, while Chinese emissions are significantly overestimated at the end of the study period. In both cases, this suggests an underestimation of the emission residuals between the global and the Chinese emissions. Consequently, GAINS also provides the lowest *a posteriori* emission estimates in almost all shown regions except China, resulting in an underestimation of the global emissions.

In the case of EDGAR, both, the *a priori* and *a posteriori* emissions agree with the reference values of the AGAGE 12-box model and NOAA growth rate emissions within 8-9%. While the *a priori* emissions are on average biased low by 6%, the *a posteriori* emissions show on average almost no bias (0.1%) compared to the reference values. We, therefore, conclude that EDGAR provides a good estimate for the accumulated $SF_6$ emissions also from poorly monitored areas, well suited for global inversions.

The average of the total global emissions of the different discussed cases provides a very good estimate for the global $SF_6$ emissions, showing an average bias of +1,4% compared to both, the AGAGE box model and the NOAA growth rate emissions,



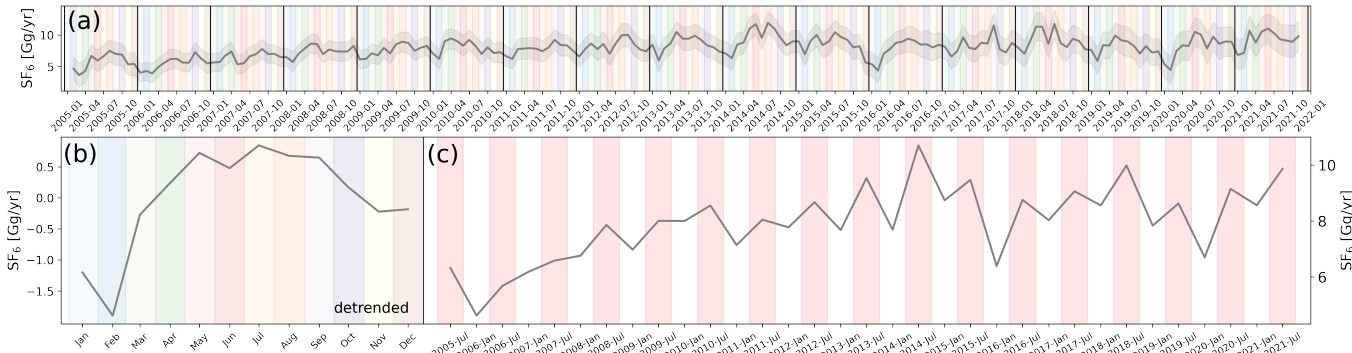

**Figure 13.** Inversion results from monthly SF$_6$ inversions for the Northern Hemisphere: (a) monthly *a posteriori* emission in the period 2005-2021, (b) detrended *a posteriori* emissions averaged for each month across all years, and (c) semi-annual *a posteriori* emissions. Distinct months are highlighted with different colors. In panel c, the specified summer (April - September) and winter periods (October - March) are shown in red and white respectively.

with an agreement within 10%. Its trend shows an increase until 2014 followed by a stabilization thereafter (similar to the Chinese emission trend). This is a pattern that can be also observed for the annual increases in the globally-averaged NOAA atmospheric SF$_6$ mole fractions, and derived emissions. Notice that the average global trend of 0.20 Gg/yr is slightly smaller than for Chinese emissions (0.21 Gg/yr), supporting the finding of An et al. (2024) that Chinese emissions alone have offset the overall decreasing emissions from all other countries.

Despite some potential problems with our inversion setup that can lead to biased *a posteriori* global emissions (as could be clearly seen and explained with the UNFCCC-ELE and GAINS *a priori* emissions), overall our *a posteriori* global emissions seem to be quite accurate, with average biases to the box model and NOAA growth rate emissions of +16%, 0.1%, and -15% for UNFCCC-ELE, EDGAR, and GAINS respectively. Even strongly biased global *a priori* emissions, as for GAINS until 2015, could be brought relatively close to the known values. This is beneficial, since our regional estimates combined are then consistent with the global emissions, which has rarely been achieved before. We attribute this capability of simultaneously constraining both regional as well as global emissions mostly to our long backward calculation period of 50 days (Vojta et al., 2022) and our extensive observation data set. However, the uncertainties of the inversion-derived emissions remain large in India and the Southern Hemisphere. While the aggregated emission in these regions is also quite well known as the residual between global emissions and emissions in well-monitored areas, the distribution of the emissions between and within these regions is less well known. Nevertheless, in most cases, the regional results at least indicate a clear direction in which *a priori* emissions need to be corrected even for these poorly monitored regions.

### 3.3.6 Seasonality of SF$_6$ emissions

Our *a priori* emission data sets contain no seasonal information and are assumed to be constant throughout the year. Figure 13 shows the monthly resolved *a posteriori* total SF$_6$ emissions in the Northern Hemisphere using the E7P *a priori* emission



inventory both for the whole time series (Fig. 13a) and as monthly averages over the whole time period, after detrending the time series (Fig. 13b) . While different years have unique seasonal patterns, a notable emission minimum can be observed at the beginning of every year (January/February) and emissions tend to be highest in the summer. This is most clearly seen in the averaged seasonal cycle (Fig. 13b), which shows a minimum in February and a broad maximum from May to September. To better demonstrate the consistency of this seasonal cycle throughout the entire period of our study, Fig. 13c shows semi-annual $SF_6$ emissions in the Northern Hemisphere, derived by averaging seasonal emissions for winter (October - March) and summer (April - September). In line with panels a) and b), Fig. 13c shows higher emissions in summer than in winter, and this pattern is found in almost every individual year.

However, the seasonal $SF_6$ emission patterns vary by region (shown for China, USA, and EU in Fig. A9). While there is no clear seasonal cycle in the EU emissions, the Chinese seasonality is similar to the one in the Northern Hemisphere (Fig. 13b). For the USA, we find an even stronger seasonal variation with a May/June peak of $SF_6$ emissions. This result is in contradiction to Hu et al. (2023), who suggested U.S. $SF_6$ emissions to peak in winter. Hu et al. (2023) argued that many U.S. companies maintain electrical equipment in the winter rather than in the summer and that cold temperatures can cause sealing materials in electrical equipment to become brittle, resulting in more leaks. We suspect that the contradictions between our two studies are mainly due to the different baseline treatments. As discussed in Sec 3.1, our baseline lowers in the summer for several stations, a feature which we argued is realistic and reflects the transport of different, cleaner air masses over the respective stations. Neglecting such a lowered baseline would lead to underestimated summer emissions. In addition, our inversion results for the USA are mainly driven by the high-frequency measurements from Trinidad Head (THD) and Niwot Ridge (NWR), which have not been used by Hu et al. (2023). A possible explanation for the summer emission maximum might be the seasonal variability of electricity generation, which peaks in summer for most of the Northern Hemisphere. In addition, the increasing $SF_6$ pressure at high summer temperatures and heat stress of the electrical equipment could lead to more leakage. However, further research on the seasonal cycle of $SF_6$ emissions is needed to provide a more conclusive answer as to the cause(s).

## 4   Conclusions

Our inversion study provides observation-based, regionally resolved global $SF_6$ emission estimates for the period 2005 - 2021, using initial conditions based on an atmospheric $SF_6$ re-analysis. We further consider different *a priori* emission inventories and use a newly compiled, extensive observation data set along with 50-day LPDM backward simulations to provide accurate estimates of the global, spatially distributed $SF_6$ emissions. Our main findings are the following:

–   The GDB approach is a robust method for estimating initial conditions, especially at challenging measurement stations. We demonstrate that it successfully accounts for meteorological variability (e.g., the Asian summer monsoon) in the baseline, reducing the need for baseline optimization by the inversion. Thus, the information content of the observations can be optimally used for improving the *a priori* emissions.





– Our inversion produces regional *a posteriori* emissions that, taken together, are consistent within 10% with the well-known global emissions based on observed atmospheric growth rates. This is a beneficial feature of our inversion setup combining accurate baselines and long (50 days) backward calculation periods.

– The global inversion shows the largest emission improvements in the high emitting regions China, USA, and Europe, where the observation networks used have good coverage. Our annual inversion results are in excellent agreement with several existing regional inversion studies focusing on these three regions.

– Annual U.S. $SF_6$ emissions strongly decreased from 1.25 Gg in 2005 to 0.48 Gg in 2021, showing a trend of -0.054 Gg/yr. However, these inversion-derived emissions are on average twice as high as the emissions reported to the UNFCCC. Thus, we find that the U.S. are systematically underreporting their $SF_6$ emissions.

– Annual total $SF_6$ emissions from EU countries show a decreasing trend of -0.006 Gg/yr, from 0.41 Gg in 2005 to 0.25 Gg in 2021. However, also Europe systematically underreports their $SF_6$ emissions to UNFCCC.

– The European emissions show a substantial drop in 2018, resulting most likely from the EU's F-gas regulation 517/2014 (European Parliament and Council of the European Union, 2014), which requires new electrical switch gear put into service from 2017 onwards to be equipped with a leak detection system and bans the use of $SF_6$ for recycling magnesium die-casting alloys from 2018. This is a good example how stringent mitigation measures can successfully reduce $SF_6$ emissions almost immediately.

– Chinese $SF_6$ emissions show an increasing trend of 0.21 Gg/yr, growing from 1.28 Gg in 2005 to 5.16 Gg in 2021, with a particularly steep trend until 2014 and a flattening afterwards. The derived trend is slightly steeper than the global total $SF_6$ emission trend (0.20 Gg/yr), supporting the suggestion that Chinese emissions alone have more than offset the overall decreasing emissions from other countries (An et al., 2024). China's official voluntary reports substantially underestimate their $SF_6$ emissions (by more than 50%).

– $SF_6$ emissions in the Southern Hemisphere and some other parts of the world (e.g., India) are hard to constrain due to insufficient coverage by observations. While the inversion most likely reduces large biases of *a priori* estimated emissions in Africa and South America, substantial uncertainties about these emissions remain. However, the EDGAR bottom-up inventory seems to provide a reliable estimate of the aggregated emissions in poorly monitored regions. Nevertheless, more observations in these regions are needed to constrain their regional distribution.

– Despite the above difficulties, our inversions suggest that India's $SF_6$ emissions have increased substantially, probably doubling since the year 2005.

– Our monthly inversion results show overall higher $SF_6$ emissions in the summer (April - September) than in winter (October - March) in the Northern Hemisphere, with a distinct minimum at the beginning of the year. While America's $SF_6$ emissions show a clear peak in May and June and China's emission pattern is similar to the Northern Hemisphere,




no clear seasonal pattern is identified for Europe. As our findings for the U.S. are in contradiction to Hu et al. (2023), we
suggest that more research on the seasonality of $SF_6$ emissions is needed.

– On the basis of the inversion results, we can neither confirm nor refute the hypothesis that the ocean sink of $SF_6$ is a
substantial part (up to 7% according to Ni et al. (2023)) of the anthropogenic emission fluxes.

– Since we find that national reports for the U.S., Europe, and China all underreport their $SF_6$ emissions, while other
countries with potentially high emissions (e.g., India) do not report their emissions at all, we suggest that bottom-up
methods to determine the emissions need to be refined. This should include a better quantification of the processes
causing the emissions that could explain the emission seasonality found here.

– Finally, countries worldwide need to reduce their emissions substantially to avoid further strong increases in the atmospheric burden of the long-lived greenhouse gas $SF_6$. The stringent regulations recently introduced in Europe are a good
example also for other countries - yet are still insufficient to stabilize the atmospheric $SF_6$ burden.



*Code and data availability.* Daily-resolved global $SF_6$ mole fraction fields between 2005 and 2021 from the global re-analysis are provided at https://doi.org/10.25365/phaidra.489. The used source code of FLEXPART 10.4 (described in detail by Pisso et al., 2019) can be found at https://doi.org/10.5281/zenodo.3542278. The used FLEXINVERT+ code (described in detail by Thompson and Stohl, 2014) together with setting files are provided at https://doi.org/10.25365/phaidra.488. The source code of FLEXPART 8-CTM-1.1 together with a user's guide can be freely downloaded at https://doi.org/10.5281/zenodo.1249190 (Henne et al., 2018). Atmospheric mole fraction measurements of $SF_6$ used

in this study are freely available from the following sources: AGAGE data: https://agage2.eas.gatech.edu/data_archive/agage/gc-ms-medusa/complete/ (Prinn et al., 2018); Heathfield Tall Tower data: https://catalogue.ceda.ac.uk/uuid/df502fe4715c4177ab5e4e367a99316b (Arnold et al., 2019); Bilsdale Tall Tower data: https://catalogue.ceda.ac.uk/uuid/d2090552c8fe4c16a2fd7d616adc2d9f (O'Doherty et al., 2019); Zeppelin mountain data: https://ebas-data.nilu.no/Pages/DataSetList.aspx?key=4548F59E3CBD48E0A505E8968BD268EB (2005-2010 EBAS, 2024); NOAA/GML Chromatograph for Atmospheric Trace Species (CATS) program: https://gml.noaa.gov/dv/data/index.php?parameter_

name=Sulfur%2BHexafluoride&type=Insitu&frequency=Hourly%2Baverages (all stations, hourly data Dutton and Hall, 2023); Monte Cimone, Cape Ochiishi, Izaña, Ragged Point, Zugspitze-Schneefernerhaus: https://gaw.kishou.go.jp/search (di Sarra et al., 2022); Atmospheric $SF_6$ Dry Air Mole Fractions from the NOAA GML Carbon Cycle Cooperative Global Air Sampling Network: https://gml.noaa.gov/aftp/data/greenhouse_gases/sf6/flask/surface/ (Lan et al., 2023; Dlugokencky et al., 2020); NOAA Global Greenhouse Gas Reference Network provided flask-air PFP sample measurements of $SF_6$ at Tall Towers and other Continental Sites https://gml.noaa.gov/aftp/data/greenhouse_gases/sf6/pfp/surface/ (Andrews et al., 2022); Atmospheric Sulfur Hexafluoride Dry Air Mole Fractions from the NOAA

GML Carbon Cycle Aircraft Vertical Profile Network https://gml.noaa.gov/aftp/data/greenhouse_gases/sf6/pfp/aircraft/: (McKain et al., 2022); NOAA ObsPACK $SF_6$ data: https://gml.noaa.gov/ccgg/obspack/data.php?id=obspack_sf6_1_v2.1_2018-07-10 (NOAA Carbon Cycle Group ObsPack Team, 2018); IAGOS-CARIBIC Aircraft measurements: https://zenodo.org/records/10495039 (Schuck and Obersteiner, 2024); NOAA/ESRL/GMD/HATS Trace Gas Measurements from Airborne Platforms: https://gml.noaa.gov/aftp/data/hats/airborne/ (Elkins

et al., 2020). For the observations at BIK (Popa et al., 2010), BRM (Rust et al., 2022), GSN (Kim et al., 2012), and HAT (Saikawa et al., 2012) we refer to E. Popa <epopa2@yahoo.com>, S. Reimann <stefan.reimann@empa.ch>, S. Park <sparky@knu.ac.kr>, and T. Saito <saito.takuya@nies.go.jp>, respectivley.

*Author contributions.* MV, AP, and AS designed the study. MV performed the FLEXPART, FLEXPART CTM, and FLEXINVERT+ simulations. RT helped with the FLEXINVERT+ setup and simulation issues. MV made the figures with contributions from SA. JM, SP and GL
provided atmospheric observation data. PP and FL provided GAINS emissions estimates. MV wrote the text with input from AP, SA, SP, GL, PP, FL, XL, JM, RT, and AS.

*Competing interests.* The authors declare that they have no conflict of interest.

*Acknowledgements.* We thank the whole AGAGE team for providing measurement data, including C.M.Harth, J. Mühle, P.K. Salameh, and R.F. Weiss (Scripps Institution of Oceanography, UCSD); P.B. Krummel, P.J. Fraser, L.P. Steele (CSIRO Environment); R.H.(J.) Wang
(Georgia Institute of Technology); S. O'Doherty, D. Young, K.M. Stanley (University of Bristol); M.K. Vollmer, S. Reimann (EMPA: Swiss



Federal Laboratories for Materials Science and Technology); C.R. Lunder, O. Hermanson (NILU: Norwegian Institute for Air Research) and particularly all station managers and station operators. AGAGE operations at Mace Head, Trinidad Head, Cape Matatula, Ragged Point, and Cape Grim are supported by NASA (USA) grants to MIT (NAG5-12669, NNX07AE89G, NNX11AF17G, NNX16AC98G) and SIO (NNX07AE87G, NNX07AF09G, NNX11AF15G, NNX11AF16G, NNX16AC96G, NNX16AC97G), and also by: Department for Business, Energy & Industrial Strategy (BEIS, UK), Contract 5488/11/2021 to the University of Bristol for Mace Head and NOAA (USA), contract 1305M319CNRMJ0028 to the University of Bristol for Ragged Point; CSIRO, BoM, DCCEEW, RRA (Australia); NILU (Norway); KNU (Korea); CMA (China); NIES (Japan); and Urbino University (Italy). For Jungfraujoch funding is acknowledged for the project HALCLIM/CLIMGAS-CH by the Swiss Federal Office for the Environment (FOEN) and for ICOS (Integrated Carbon Observation System) by the Swiss National Science Foundation. In addition, measurements are supported by the International Foundation High Altitude Research Stations Jungfraujoch and Gornergrat (HFSJG). The Commonwealth Scientific and Industrial Research Organisation (CSIRO, Australia) and Bureau of Meteorology (Australia) are thanked for their ongoing long-term support and funding of the Cape Grim station and the Cape Grim science program. Operations of the Gosan station on Jeju Island, South Korea were supported by the National Research Foundation of Korea grant funded by the Korean government MSIT no. RS-2023-00229318. We also thank the NOAA Global Monitoring Laboratory for providing access to their data, including G. Dutton, J.W. Elkins, F. Moore, E. Hintsa, D. Hurst, B. Hall, K. McKain, S. Montzka, B. Miller, C. Sweeney, E. Dlugokencky, A. Andrews, D. Nance, C. M. Harth, and key partners: L. Huang (EC), K. Davis (PSU), Observations collected in the Southern Great Plains (SGP) by S. Biraud (Lawrence Berkeley National Laboratory, USA) were supported by the Office of Biological and Environmental Research of the US Department of Energy under contract DE-AC02-05CH11231 as part of the Atmospheric Radiation Measurement Program (ARM) and Environmental System Science (ESS) Program.

We further acknowledge the following people and institutions for sharing their observation data: T. Saito (National Institute for Environmental Studies, Japan); E. Cuevas (State Meteorological Agency, Spain); D. Say, (University of Bristol), J. Arduini (University of Urbino), C. Couret (German Environment Agency), Elena Popa (Utrecht University), S. Sugawara (Miyagi University of Education, Japan), A. Rauthe, J. Williams (Max Planck Institute for Chemistry), T. Schuck (Goethe University Frankfurt), F. Obersteiner (Karlsruhe Institute of Technology), C. Couret (German Federal Environmental Agency), L. Gatti (Instituto Nacional de Pesquisas Espaciais).

The computational results presented have been achieved in part using the Vienna Scientific Cluster (VSC), Project 71878: Demonstration of a Lagrangian re-analysis, to which we are grateful. Further acknowledgment is made for the use of ECMWF's computing and archive facilities provided through a special project (spatvojt) in this research. The study was supported by EYE-CLIMA, a European Union's Horizon Europe research and innovation programme under grant agreement No. 101081395. We thank L. Hu and C. Groot Zwaaftink for the discussions. We further thank M. Dütsch, L. Bakels, S. Bucci, K. Baier, D. Tatsii, An. Raju, R. Subramanian, S. Wittig, O. Nabavi, I. Evangelou, M. Blaschek, A. Skorokhod, P. Seibert (University of Vienna) for their support.



**Table A1.** Inversion results for the annual SF$_6$ emissions from the United States of America in the period 2005-2021. Annual emissions are shown together with their 1-$\sigma$ uncertainties, for different *a priori* emissions inventories. We also provide an average of the inversion results, while respective uncertainties represent the minimum and maximum uncertainty limits across the results.

| annual total SF$_6$ emissions from the United States of America | | | |
|---|---|---|---|
| year | UNFCCC-ELE [Gg/yr] | EDGAR [Gg/yr] | GAINS [Gg/yr] | average [Gg/yr] |
| 2005 | 1.21 ± 0.16 | 1.44 ± 0.34 | 1.11 ± 0.11 | 1.25 [1.00, 1.78] |
| 2006 | 1.34 ± 0.15 | 1.58 ± 0.32 | 1.24 ± 0.10 | 1.38 [1.14, 1.90] |
| 2007 | 1.14 ± 0.13 | 1.38 ± 0.30 | 1.04 ± 0.09 | 1.19 [0.95, 1.67] |
| 2008 | 1.24 ± 0.12 | 1.51 ± 0.27 | 1.09 ± 0.08 | 1.28 [1.01, 1.78] |
| 2009 | 0.99 ± 0.10 | 1.16 ± 0.26 | 0.93 ± 0.08 | 1.03 [0.86, 1.42] |
| 2010 | 0.86 ± 0.10 | 1.13 ± 0.27 | 0.80 ± 0.07 | 0.93 [0.73, 1.39] |
| 2011 | 0.75 ± 0.10 | 1.00 ± 0.27 | 0.68 ± 0.07 | 0.81 [0.61, 1.27] |
| 2012 | 0.62 ± 0.09 | 0.90 ± 0.28 | 0.57 ± 0.08 | 0.70 [0.50, 1.18] |
| 2013 | 0.58 ± 0.09 | 0.84 ± 0.27 | 0.51 ± 0.07 | 0.64 [0.44, 1.11] |
| 2014 | 0.52 ± 0.08 | 0.63 ± 0.27 | 0.44 ± 0.06 | 0.53 [0.36, 0.90] |
| 2015 | 0.52 ± 0.08 | 0.67 ± 0.25 | 0.45 ± 0.06 | 0.55 [0.38, 0.92] |
| 2016 | 0.72 ± 0.08 | 0.96 ± 0.24 | 0.63 ± 0.06 | 0.77 [0.57, 1.21] |
| 2017 | 0.58 ± 0.08 | 0.80 ± 0.24 | 0.52 ± 0.06 | 0.63 [0.45, 1.04] |
| 2018 | 0.55 ± 0.08 | 0.80 ± 0.25 | 0.49 ± 0.06 | 0.61 [0.42, 1.05] |
| 2019 | 0.53 ± 0.08 | 0.75 ± 0.26 | 0.46 ± 0.06 | 0.58 [0.40, 1.02] |
| 2020 | 0.44 ± 0.08 | 0.64 ± 0.25 | 0.39 ± 0.06 | 0.49 [0.32, 0.89] |
| 2021 | 0.45 ± 0.09 | 0.62 ± 0.27 | 0.38 ± 0.06 | 0.48 [0.32, 0.89] |



**Table A2.** Inversion results for the annual total SF$_6$ emissions from EU countries in the period 2005-2021. Annual emissions are shown together with their 1-$\sigma$ uncertainties, for different *a priori* emissions inventories. We also provide an average of the inversion results, while respective uncertainties represent the minimum and maximum uncertainty limits across the results.

| annual total SF$_6$emissions from EU countries | | | |
|---|---|---|---|
| **year** | **UNFCCC-ELE** [Gg/yr] | **EDGAR** [Gg/yr] | **GAINS** [Gg/yr] | **average** [Gg/yr] |
| 2005 | $0.43 \pm 0.09$ | $0.38 \pm 0.09$ | $0.43 \pm 0.08$ | 0.41 [0.29, 0.52] |
| 2006 | $0.39 \pm 0.09$ | $0.35 \pm 0.08$ | $0.39 \pm 0.07$ | 0.38 [0.27, 0.47] |
| 2007 | $0.35 \pm 0.08$ | $0.34 \pm 0.08$ | $0.36 \pm 0.07$ | 0.35 [0.26, 0.44] |
| 2008 | $0.37 \pm 0.08$ | $0.35 \pm 0.08$ | $0.36 \pm 0.07$ | 0.36 [0.27, 0.46] |
| 2009 | $0.37 \pm 0.09$ | $0.34 \pm 0.09$ | $0.38 \pm 0.08$ | 0.36 [0.25, 0.46] |
| 2010 | $0.40 \pm 0.08$ | $0.37 \pm 0.08$ | $0.40 \pm 0.07$ | 0.39 [0.29, 0.48] |
| 2011 | $0.36 \pm 0.08$ | $0.30 \pm 0.08$ | $0.35 \pm 0.08$ | 0.34 [0.22, 0.44] |
| 2012 | $0.38 \pm 0.08$ | $0.37 \pm 0.08$ | $0.40 \pm 0.07$ | 0.38 [0.29, 0.48] |
| 2013 | $0.29 \pm 0.07$ | $0.29 \pm 0.07$ | $0.32 \pm 0.07$ | 0.30 [0.22, 0.39] |
| 2014 | $0.33 \pm 0.07$ | $0.32 \pm 0.07$ | $0.36 \pm 0.06$ | 0.34 [0.26, 0.42] |
| 2015 | $0.31 \pm 0.07$ | $0.31 \pm 0.08$ | $0.32 \pm 0.07$ | 0.31 [0.23, 0.39] |
| 2016 | $0.37 \pm 0.07$ | $0.35 \pm 0.07$ | $0.37 \pm 0.07$ | 0.36 [0.27, 0.44] |
| 2017 | $0.39 \pm 0.08$ | $0.37 \pm 0.08$ | $0.39 \pm 0.07$ | 0.38 [0.29, 0.47] |
| 2018 | $0.26 \pm 0.07$ | $0.25 \pm 0.08$ | $0.28 \pm 0.06$ | 0.26 [0.18, 0.34] |
| 2019 | $0.28 \pm 0.07$ | $0.27 \pm 0.07$ | $0.30 \pm 0.06$ | 0.28 [0.20, 0.37] |
| 2020 | $0.32 \pm 0.06$ | $0.32 \pm 0.07$ | $0.34 \pm 0.06$ | 0.33 [0.25, 0.40] |
| 2021 | $0.25 \pm 0.06$ | $0.25 \pm 0.08$ | $0.26 \pm 0.07$ | 0.25 [0.16, 0.33] |



**Table A3.** Inversion results for the annual Chinese SF$_6$ emissions in the period 2005-2021. Annual emissions are shown together with their 1-$\sigma$ uncertainties, for different *a priori* emissions inventories. We also provide an average of the inversion results, and an average excluding the GAINS-derived inversion, while uncertainties represent the minimum and maximum uncertainty limits across the respective inversions results.

| annual total SF$_6$emissions from China | | | | | |
|---|---|---|---|---|---|
| year | **UNFCCC-ELE** [Gg/yr] | **EDGAR** [Gg/yr] | **GAINS** [Gg/yr] | **average** [Gg/yr] | **average without GAINS** [Gg/yr] |
| 2005 | $1.35 \pm 0.45$ | $1.21 \pm 0.32$ | $2.13 \pm 0.31$ | 1.56 [0.89, 2.44] | 1.28 [0.89, 1.80] |
| 2006 | $1.14 \pm 0.36$ | $1.08 \pm 0.33$ | $1.78 \pm 0.42$ | 1.33 [0.75, 2.20] | 1.11 [0.75, 1.50] |
| 2007 | $2.33 \pm 0.51$ | $2.26 \pm 0.37$ | $2.97 \pm 0.58$ | 2.52 [1.82, 3.55] | 2.29 [1.82, 2.84] |
| 2008 | $2.78 \pm 0.51$ | $2.72 \pm 0.37$ | $3.53 \pm 0.68$ | 3.01 [2.27, 4.22] | 2.75 [2.27, 3.29] |
| 2009 | $3.24 \pm 0.59$ | $3.17 \pm 0.42$ | $3.84 \pm 0.80$ | 3.42 [2.66, 4.64] | 3.21 [2.66, 3.83] |
| 2010 | $3.13 \pm 0.55$ | $3.12 \pm 0.46$ | $3.99 \pm 0.87$ | 3.41 [2.59, 4.86] | 3.13 [2.59, 3.68] |
| 2011 | $2.91 \pm 0.60$ | $2.95 \pm 0.50$ | $3.81 \pm 0.97$ | 3.22 [2.32, 4.78] | 2.93 [2.32, 3.51] |
| 2012 | $3.44 \pm 0.62$ | $3.50 \pm 0.54$ | $4.27 \pm 1.10$ | 3.73 [2.82, 5.37] | 3.47 [2.82, 4.05] |
| 2013 | $4.14 \pm 0.69$ | $4.19 \pm 0.58$ | $5.42 \pm 1.21$ | 4.59 [3.45, 6.64] | 4.17 [3.45, 4.84] |
| 2014 | $4.89 \pm 0.69$ | $4.96 \pm 0.60$ | $6.09 \pm 1.32$ | 5.31 [4.20, 7.41] | 4.92 [4.20, 5.57] |
| 2015 | $4.53 \pm 0.73$ | $4.61 \pm 0.65$ | $5.96 \pm 1.38$ | 5.03 [3.80, 7.34] | 4.57 [3.80, 5.26] |
| 2016 | $3.60 \pm 0.80$ | $3.57 \pm 0.71$ | $4.37 \pm 1.58$ | 3.85 [2.79, 5.95] | 3.58 [2.79, 4.40] |
| 2017 | $4.10 \pm 0.81$ | $4.15 \pm 0.74$ | $5.21 \pm 1.60$ | 4.49 [3.29, 6.82] | 4.12 [3.29, 4.90] |
| 2018 | $4.72 \pm 0.86$ | $4.82 \pm 0.79$ | $6.02 \pm 1.67$ | 5.18 [3.86, 7.70] | 4.77 [3.86, 5.60] |
| 2019 | $3.99 \pm 0.85$ | $4.00 \pm 0.79$ | $4.85 \pm 1.72$ | 4.28 [3.13, 6.57] | 3.99 [3.13, 4.84] |
| 2020 | $4.38 \pm 0.96$ | $4.48 \pm 0.89$ | $5.27 \pm 1.86$ | 4.71 [3.41, 7.13] | 4.43 [3.42, 5.38] |
| 2021 | $5.12 \pm 1.00$ | $5.21 \pm 0.94$ | $6.41 \pm 2.09$ | 5.58 [4.12, 8.50] | 5.16 [4.12, 6.15] |




**Table A4.** Inversion results for the annual global total SF$_6$ emissions in the period 2005-2021. Annual emissions are shown together with their 1-$\sigma$ uncertainties, for different *a priori* emissions inventories. We also provide an average of the inversion results, while respective uncertainties represent the minimum and maximum uncertainty limits across the results.

| Annual global total SF$_6$emissions | | | |
|---|---|---|---|
| year | **UNFCCC-ELE** [Gg/yr] | **EDGAR** [Gg/yr] | **GAINS** [Gg/yr] | **average** [Gg/yr] |
| 2005 | 6.41 ± 1.96 | 5.59 ± 1.63 | 4.54 ± 0.73 | 5.51 [3.81, 8.37] |
| 2006 | 5.97 ± 1.92 | 5.38 ± 1.33 | 4.52 ± 0.83 | 5.29 [3.69, 7.89] |
| 2007 | 8.23 ± 2.00 | 6.54 ± 1.34 | 5.38 ± 0.98 | 6.72 [4.40, 10.23] |
| 2008 | 9.56 ± 1.97 | 7.89 ± 1.31 | 6.63 ± 1.06 | 8.02 [5.57, 11.53] |
| 2009 | 8.88 ± 1.97 | 7.77 ± 1.37 | 6.68 ± 1.19 | 7.78 [5.48, 10.85] |
| 2010 | 9.94 ± 2.01 | 8.26 ± 1.41 | 6.91 ± 1.25 | 8.37 [5.66, 11.95] |
| 2011 | 9.42 ± 2.06 | 7.85 ± 1.44 | 6.41 ± 1.35 | 7.90 [5.06, 11.48] |
| 2012 | 10.54 ± 2.17 | 8.39 ± 1.53 | 6.97 ± 1.48 | 8.63 [5.49, 12.71] |
| 2013 | 10.72 ± 2.20 | 8.88 ± 1.58 | 7.93 ± 1.60 | 9.18 [6.34, 12.92] |
| 2014 | 11.96 ± 2.27 | 9.99 ± 1.56 | 8.83 ± 1.68 | 10.26 [7.15, 14.23] |
| 2015 | 11.31 ± 2.21 | 9.39 ± 1.62 | 8.41 ± 1.75 | 9.70 [6.65, 13.52] |
| 2016 | 8.72 ± 2.30 | 7.79 ± 1.66 | 6.89 ± 1.94 | 7.80 [4.95, 11.02] |
| 2017 | 10.14 ± 2.37 | 8.56 ± 1.71 | 7.46 ± 1.98 | 8.72 [5.48, 12.50] |
| 2018 | 11.48 ± 2.34 | 9.88 ± 1.79 | 8.31 ± 2.04 | 9.89 [6.27, 13.82] |
| 2019 | 9.15 ± 2.41 | 8.02 ± 1.81 | 7.02 ± 2.11 | 8.06 [4.91, 11.55] |
| 2020 | 9.81 ± 2.50 | 8.23 ± 1.91 | 7.29 ± 2.24 | 8.44 [5.05, 12.31] |
| 2021 | 11.11 ± 2.56 | 9.50 ± 2.03 | 8.41 ± 2.50 | 9.67 [5.91, 13.67] |



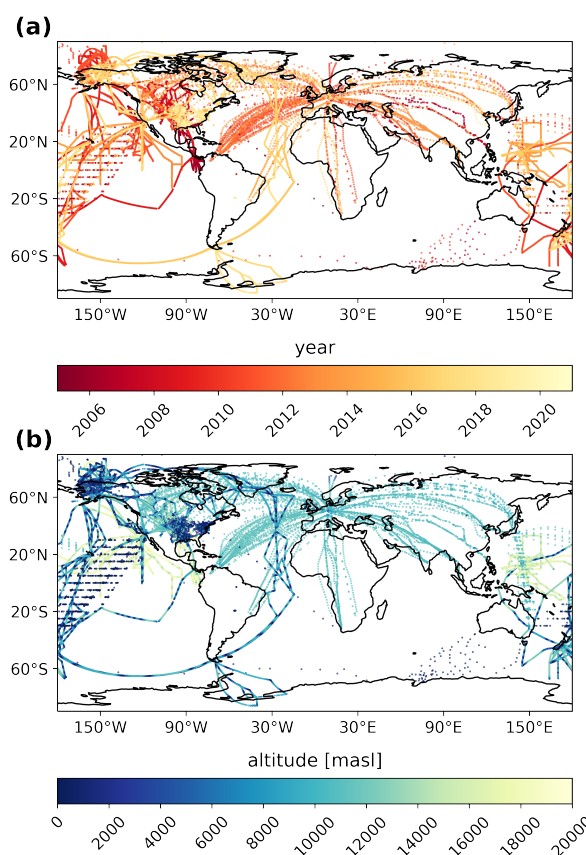

**Figure A1.** Observations from aircraft and ship campaigns from 2005 - 2021. The color bars indicate (a) the measurement date and (b) the altitude of the respective observations.



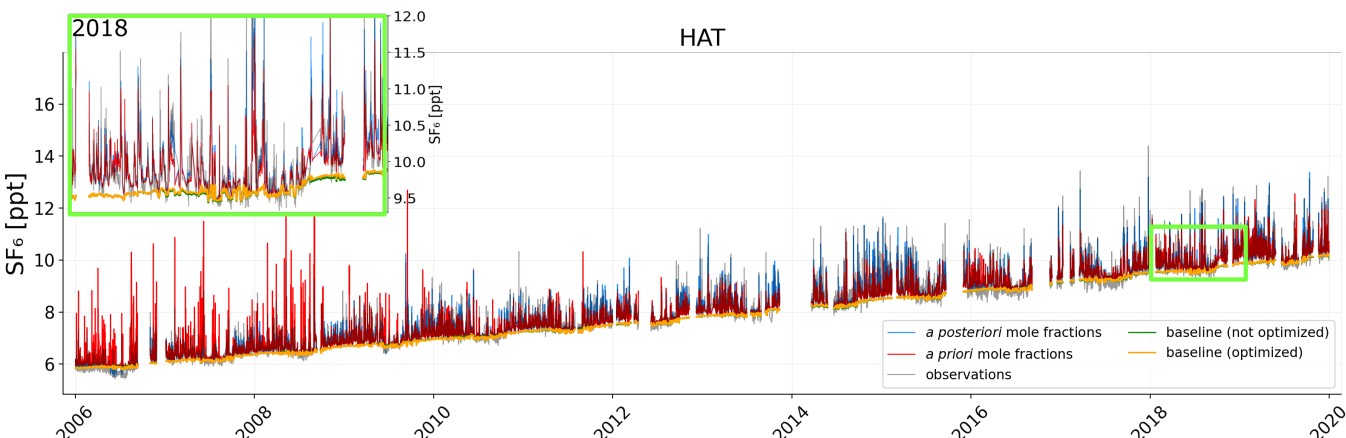

**Figure A2.** Mole fraction time series at the Hateruma (Japan) measurement station. Red lines represent the modeled *a priori* mole fractions calculated with the UP *a priori* emissions and blue lines represent the modeled *a posteriori* mole fractions. The green line illustrates the baseline derived by the GDB method and the orange line shows the optimized baseline. The grey line represents the observed mole fractions. The inset panel zooms into the year 2018, as illustrated by the lightgreen rectangle.

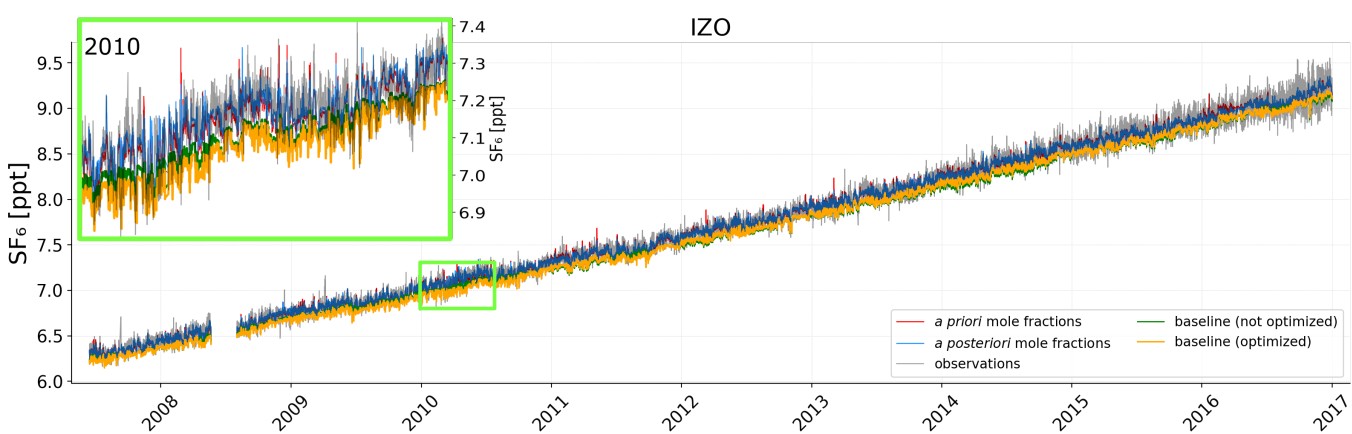

**Figure A3.** Mole fraction time series at the Izaña (Tenerife) measurement station. Red lines represent the modeled *a priori* mole fractions calculated with the UP *a priori* emissions and blue lines represent the modeled *a posteriori* mole fractions. The green line illustrates the baseline derived by the GDB method and the orange line shows the optimized baseline. The grey line represents the observed mole fractions. The inset panel zooms into the year 2010, as illustrated by the lightgreen rectangle.



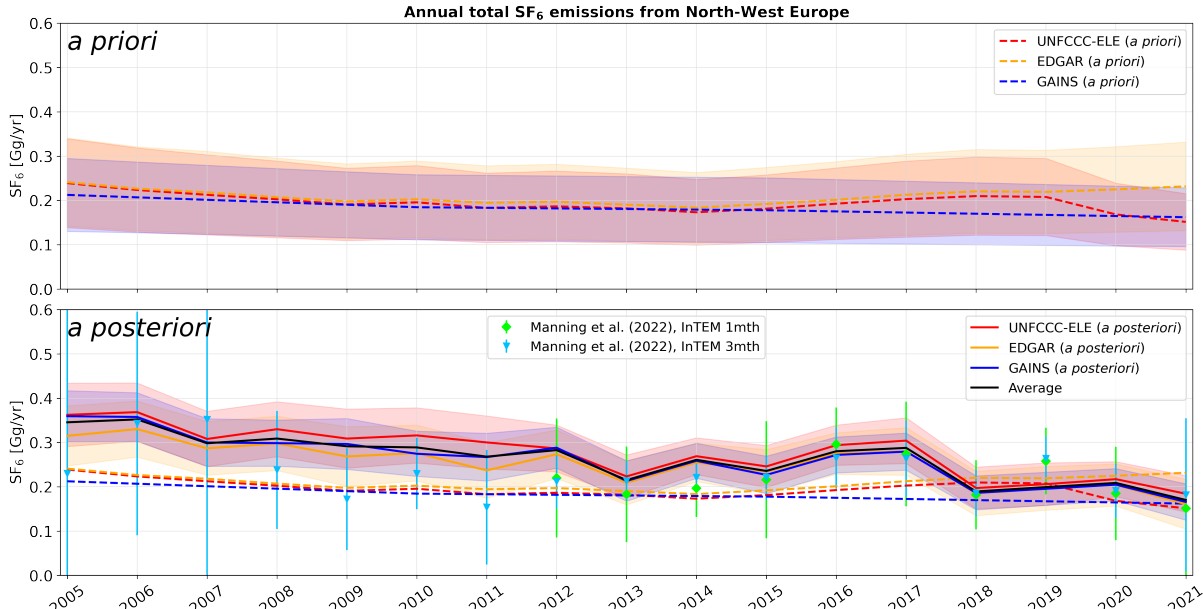

**Figure A4.** Annual *a priori* (dashed lines) and *a posteriori* (solid lines) SF$_6$ emissions from North-West Europe, shown for the period between 2005 and 2021 when using different *a priori* emissions (UNFCCC-ELE red, EDGAR orange, GAINS blue). The *a priori* emissions (top panel) and *a posteriori* emissions (bottom panel) are shown together with their respective 1-$\sigma$ uncertainties (colored shadings). The bottom panel also shows the average *a posteriori* emissions with a black solid line. The blue rectangles and the green diamonds represent the results from Manning et al. (2022) using the InTEM (Inversion Technique for Emissions Modelling) model with inversion time frames set to 3- and 1-months, respectively. For better comparison, the *a priori* emissions (without uncertainties) are also included in the bottom panel.



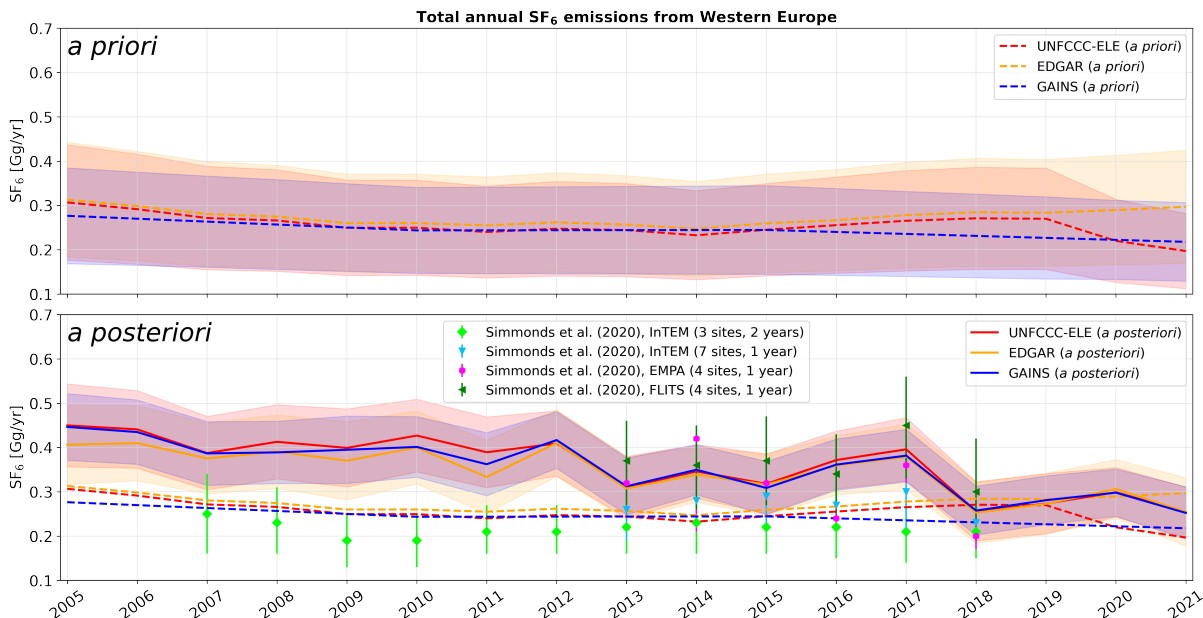

**Figure A5.** Annual *a priori* (dashed lines) and *a posteriori* (solid lines) SF$_6$ emissions from Western Europe, shown for the period between 2005 and 2021 when using different *a priori* emissions (UNFCCC-ELE red, EDGAR orange, GAINS blue). The *a priori* emissions (top panel) and *a posteriori* emissions (bottom panel) are shown together with their respective 1-$\sigma$ uncertainties (colored shadings). The bottom panel also shows the average *a posteriori* emissions (black solid line), together with the results from Simmonds et al. (2020), using four different inversion setups.

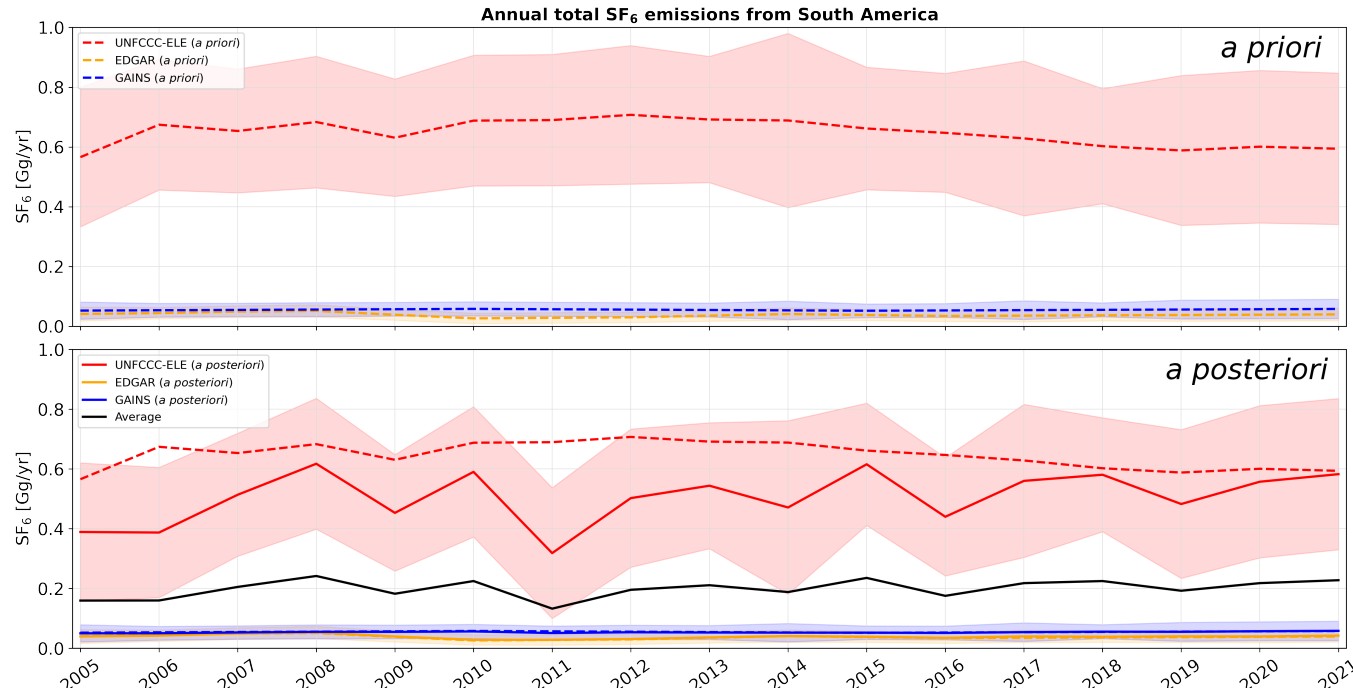

**Figure A6.** Annual *a priori* (dashed lines) and *a posteriori* (solid lines) SF$_6$ emissions from South America, shown for the period between 2005 and 2021 when using different *a priori* emissions (UNFCCC-ELE red, EDGAR orange, GAINS blue). The *a priori* emissions (top panel) and *a posteriori* emissions (bottom panel) are shown together with their respective 1-$\sigma$ uncertainties (colored shadings). The bottom panel also shows the average *a posteriori* emissions with a black solid line. For better comparison, the *a priori* emissions (without uncertainties) are also included in the bottom panel.





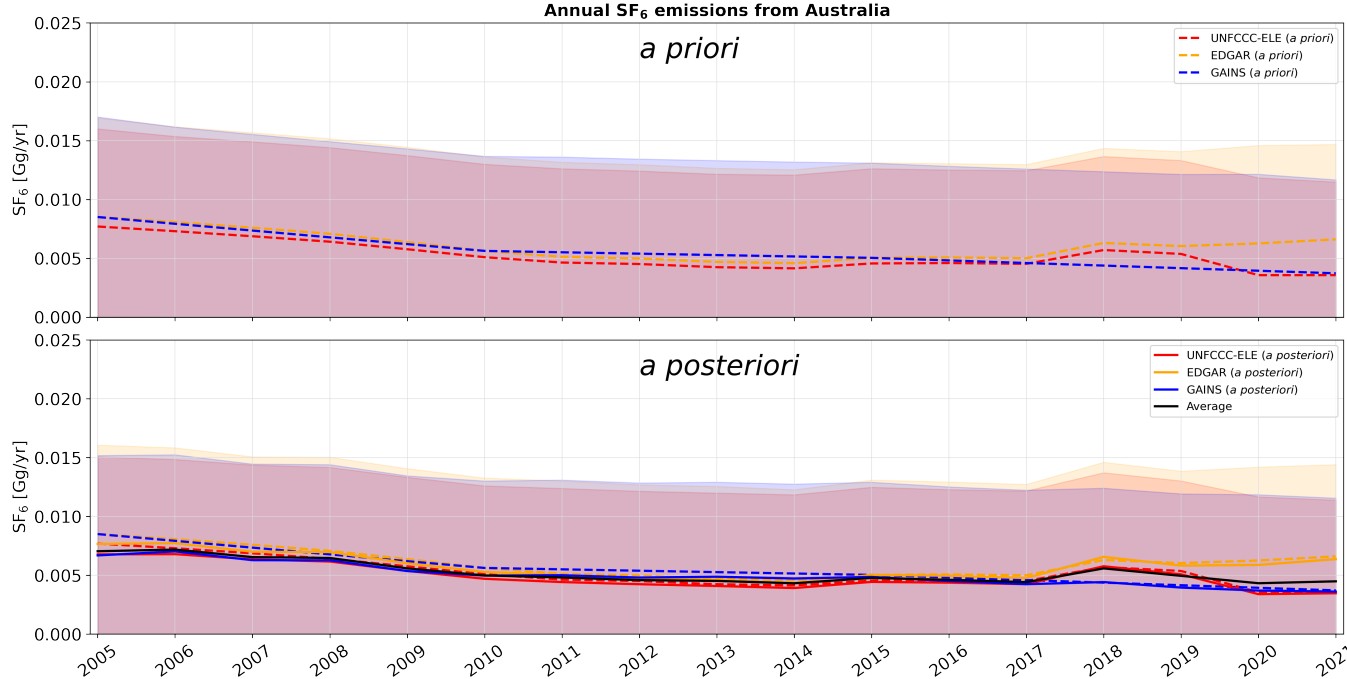

**Figure A7.** Annual *a priori* (dashed lines) and *a posteriori* (solid lines) SF$_6$ emissions from Australia, shown for the period between 2005 and 2021 when using different *a priori* emissions (UNFCCC-ELE red, EDGAR orange, GAINS blue). The *a priori* emissions (top panel) and *a posteriori* emissions (bottom panel) are shown together with their respective 1-$\sigma$ uncertainties (colored shadings). The bottom panel also shows the average *a posteriori* emissions with a black solid line. For better comparison, the *a priori* emissions (without uncertainties) are also included in the bottom panel.

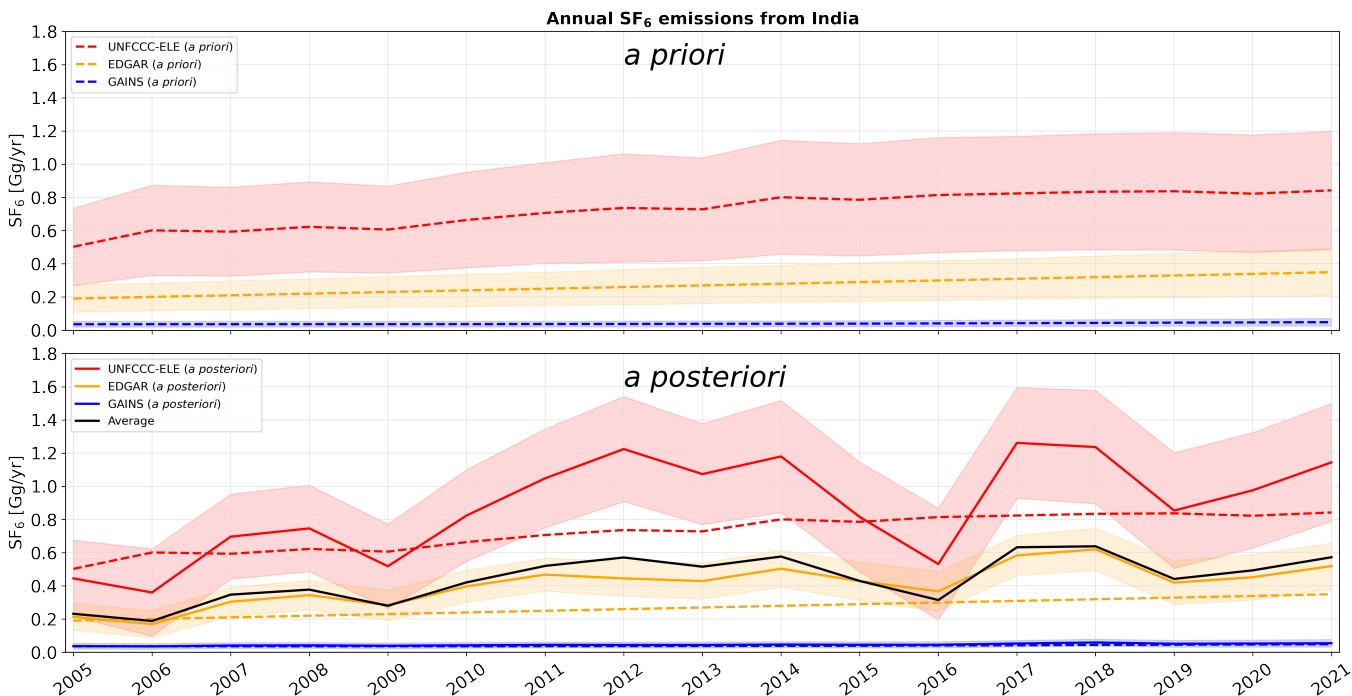

**Figure A8.** Annual *a priori* (dashed lines) and *a posteriori* (solid lines) SF$_6$ emissions from India, shown for the period between 2005 and 2021 when using different *a priori* emissions (UNFCCC-ELE red, EDGAR orange, GAINS blue). The *a priori* emissions (top panel) and *a posteriori* emissions (bottom panel) are shown together with their respective 1-$\sigma$ uncertainties (colored shadings). The bottom panel also shows the average *a posteriori* emissions with a black solid line. For better comparison, the *a priori* emissions (without uncertainties) are also included in the bottom panel.





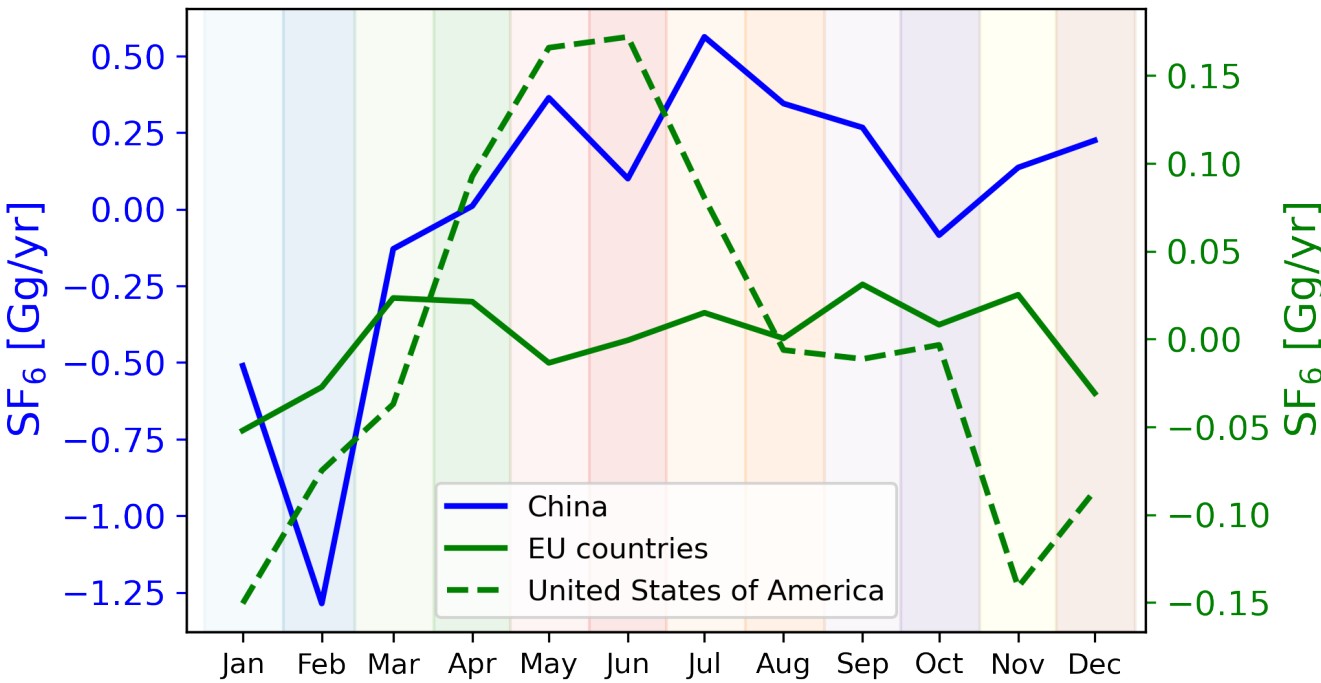

**Figure A9.** Seasonal variation of SF$_6$ emissions in China, the United States of America, and EU countries. The figure shows detrended monthly inversion results averaged for each month across all years in the period 2005-2021



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
