# Peer review of "A global re-analysis of regionally resolved emissions and atmospheric mole fractions of SF6 for the period 2005-2021"

_EGUsphere, 2024_

## Author Comment (AC1)

**Author's response #1**

This paper presents a comprehensive understanding of $SF_6$ emissions worldwide using inverse modeling, which improves our knowledge on global SF6 emissions and their regional (spatial) distributions. It could be the first study that I am aware of presenting SF6 emissions worldwide from an extensive combination of measurements. The manuscript is well-written and well-structured. The analyses are reasonable and conclusions are generally solid. The description of the methods is generally clear. The authors have also acknowledged the uncertainties of the derived emissions and the limitations of the current observation in constraining emissions in several regions including South America etc., and call for attention for enhancing measurement network in these regions, which is important for not only SF6 but others important gases.

The authors have done a very nice piece of work, that will be of interest to the community! I would recommend the publication of this manuscript on ACP. The following are some specific/minor comments, corrections or questions:

We would like to thank reviewer #1 for the valuable and constructive review of our manuscript. The suggestions for improvements were very helpful and we incorporated almost all of them into the final version of the manuscript.

In the response we use 4 different colors. The blue-colored text is the general answer to the reviewer's comments. Additionally, we show how the text is changed in the manuscript: The original text is colored grey, removed text is colored red, and new text is colored green.

Line 15: I suggest making the potentially increasing emissions in countries other than China (e.g. India) a separate point in the abstract, together with the accompanied uncertainties (arising from the limited measurement sites) which you already stated in point(5). The emissions in these countries are very important for understanding the global SF6 emissions and their variations.

We added:

(5) Our inversions indicate increasing emissions in poorly monitored areas (e.g. India, Africa, South America), however, these results are uncertain due to weak observational constraints, highlighting the need for enhanced monitoring in these areas.

Line 30: I suggest changing concentrations to mole fractions. Concentration is more for something per volume.

done

Line 48: Typo "shwn" to "shown"

done

Line 49: It should be top-down, not bottom-up

done

Line 100: how do you chose the "3h interval"? Auto-correlated errors could still exist within this period (biased systematic errors). Have you tested different intervals?

The choice of this 3h interval is, of course, subjective and can be debated. The choice is always a compromise between getting the most information out of the observation network while trying to limit the effect of error correlations.  One option would be a pre-selection of observations in addition to 3h-averaging intervals, (e.g. using only nighttime measurements for mountain stations and afternoon measurements for others, or basing the selection on meteorological conditions (e.g. Lunt et al., 2021)). We also thought about using 1-day averages, but in the end, we decided to stay with 3h intervals for this study, mostly due to the generally limited number of $SF_6$ observations (especially at the beginning of the study period) and the concerns of discarding too much information.

Line 111: I would like to know more about this 50-day back-trajectory duration. Why do you choose 50 days? Will the uncertainties from the Lagrangian dispersion model increase rapidly when running time grows? I believe it is not a primary task of this study but I suggest adding a brief explanation of the period chosen here in the Method section. You give some details in Lines 261-265 but I do not think the Figure 5b is enough to illustrate your statement in the text.

Certainly! The choice of the backward simulation period was motivated by our last study (Vojta et al. 2022), where we tested the effect of different simulation periods (1-50 days). On one hand side, we found that 50 day-simulation periods resulted in the best model-measurement agreement (e.g. correlation, MSE, Bias). Further, inversion results became less sensitive to large biases in the a priori emissions, when increasing the simulation period from 10 or 20 to 50 days.  On the other hand side, observations are most sensitive to emissions occurring during the first few days of the backward simulation and the spatially resolved information content decreases with temporal distance to the measurements, as virtual particles are spread over larger areas. While, with longer simulation periods, more emissions become accessible to the inversion, it also becomes more difficult to extract information on individual emission sources. The benefit obtained from every additional simulation day will, therefore, typically decrease, while computational costs grow. The error of individual trajectories will also grow, however, as simultaneously also the retro-plume grows, we think that the error of single trajectories becomes less and less important (due to the statistical approach of FLEXPART looking at average residence times rather than the individual trajectories). However, if longer model runs result in biases (e.g. unphysical accumulation of particles) this will set a further limit to the simulation period, in addition to the computational resources.  Therefore, we also didn't want to exceed

the tested 50 day period and additionally, computational costs of the LPDM runs for the used extensive dataset were already very high.

About the lines 261-265, We agree. We still think, it is very nice to see, that with 50-day simulation periods, there are increments over the baseline caused by emissions within the last 50 days, even at a remote station like Ragged Point and the optimization of these emission contributions can be seen. However, it is of course true that we don't show the situation for shorter periods. For this, we refer the read to Vojta et al. (2022).

We added:

The choice of the 50-day simulation period was motivated by the findings of Vojta et al. (2022) who tested the effect of different simulation periods (1-50 days), and found that 50-day simulations resulted in an improved model-measurement agreement and in more robust inversion results in comparison to shorter periods (e.g. 1, 5, 10, or 20 days).

And removed:

"Figure 5b further illustrates the advantage of choosing a rather long 50-day backward simulation period. With this long simulation period, we can see that this remote station is also directly influenced by emissions (i.e., enhancements over the baseline) that can be directly optimized. With shorter simulation times (e.g., 5-10 days), no emission contributions above the baseline could be seen, thus rendering this station useless for emission optimization. For a detailed discussion about the LPDM backward simulation period see Vojta et al. 2022}."

Line 119: I suggest adding the observation error term to eq (1).

In this case we would like to keep it as it is, as y represents the modeled mole fractions

Line 133: I suggest changing "cannot be determined well" to "may not be determined well". You need to combine with your error reductions to decide whether these emissions can be constrained well, which I suggest adding in the results and discussion section.

done

Line 157: Here you are using the "UP" a priori emissions to drive the baselines. What is the reason for choosing this? You stated later that you cannot determine which prior is better (Line 239-240), but eventually find in line 448 that EDGAR is actually the most

reliable estimate for SF6 emissions. Why not using EDGAR in generating the mole fraction field?

Yes, we concluded that EDGAR provides the best estimate for the total sum of all the emissions in poorly covered regions. On the other hand, we also saw that EDGAR is overestimating emissions in the U.S., so the choice is not totally clear and dependent on the region. Overall, indeed, EDGAR might be better suited for generating the mole fraction field. However, that is an outcome of our study and we didn't have this information when we produced the mole fraction fields.  We also thought it wouldn't be right to use this outcome a priori and re-calculate the mole fraction fields (apart from the very high computation costs), since this would mean to use a posteriori information as an input to the inversion. Further, tests showed, that the produced 3-d fields were not very sensitive to the emission inventory used.  The comparison between the produced mole fraction fields and the measurements showed a very good agreement (also for observations not used for nudging), so we were quite happy with the produced fields.

Line 219: It is good to show the available number of observations each year.

We added: …. "ranging from a minimum of 5841 (2005) to a maximum of 11901 (2016), which is related to the number of available observations each year (see Fig. S1)"

Line 226: Eq (3), I would like to know whether you apply the same inequality constraints for a posteriori baselines? Give more details of how the baselines, posterior error matrix for emission and baseline, prior and posterior uncertainty for y (observation error matrix) etc. are determined or calculated.

No, the inequality constraint was only applied to the a posteriori emissions:

Certainly, we added:

We use the analytic solution to minimize $\mathbf{J}$, which reads:

$$\hat{\mathbf{x}} = \mathbf{x}_p + \mathbf{G}(\mathbf{y} - \mathbf{H}\mathbf{x}_p)$$

with the defined Gain matrix $\mathbf{G}$:

$$\mathbf{G} = \mathbf{B}\mathbf{H}^T(\mathbf{H}\mathbf{B}\mathbf{H}^T + \mathbf{R})^{-1}$$

and:

"FLEXINVERT+ assumes a diagonal observation error covariance matrix R, and therefore, does not account for possible error correlations. The diagonal elements represent the sum of measurement and model errors, where we assume the latter to be dominant. Our error estimates are based on a number of initial inversion runs, where we assessed the model error according to the a posteriori model residuals

(difference between observed and a posteriori simulated mole fractions), and such that the reduced chi-square value (the value of the cost function at minimum divided by the number of observations and divided by 2) is close to 1. The a posteriori emission error covariance matrix **B** is calculated as

$$\hat{B} = B - GHB$$

Line 230ff: Do you show the results of the sensitivity tests somewhere? I would suggest doing so in Supplement.

We added a section about sensitivity tests in the supplements and added:

(see Sec. S7)

Line 243: suggest revising it to "observed and modeled mole fractions (before and after the inversion) at the Gosan observation station....., using the E7P emissions field as the a priori in the inversion", to make it clear.

done

Line 259: define "detrended" here when it first appears.

We added:

(detrended; i.e. removing the 2005-2021 trend from the time series)

Line 268: add how you calculate the uncertainty reductions in the Methods section. It will be easier for readers without inversion expertise. Do you use the Averaging Kernal? Will the posterior uncertainties the same with analytical solution when no inequality constraints are used?

We added to the methods section:

The relative uncertainty reduction was calculated for every grid cell, based on the *a priori* and *a posteriori* emission uncertainties in the respective cell as: $1 - \frac{a\ posteriori\ uncertainty}{a\ priori\ uncertainty}$.

The inequality constraint only affected the *a posteriori* emissions but had generally only little effect on the inversion results. The a posteriori uncertainties were not affected.

Figure 6: I am curious about the increments in the very northeast region of China. In the prior distribution, emission is already very high in this region. Is there any explanation for this? In addition, you are using the average of different variations of a priori as the inversion results, but here you are showing the increments and uncertainty reductions for only one of the variation. Suggest showing the average here, or show plots for all the variations in a separate supplement file.

Yes, the values in the very northeast region of China appear very high due to the logarithmic color bar. We also thought about using linear color bars, as this might be favorable for individual countries (such as China), however, for the global emission distribution one needs a logarithmic scale, so we decided to go with lognormal color bars. When we use a linear color bar for the Chinese a priori emissions and increments comparable to e.g. An et al. 2024, those high values "disappear", and emissions appear in a similar (maybe more familiar) pattern. However, in case of the GAINS inventory higher positive increments can be seen in the northeast region compared to EDGAR or UNFCCC-ELE.

[Figure]

We included the 3 remaining a priori emission fields, increments, error reductions and all 6 *a posteriori* emission fields in the supplements.

Figure 7: I would like to see the separate posteriori emission spatial distributions from 6 different prior distributions. Also, in Line 235-236, you claim that inversion results using different variations are similar, I suggest showing them in SI Figure.

Done – see last comment

Line 304: I hope you can discuss a bit about the interannual variations in the posterior emissions, e.g., the increase in 2019-2020 then drop. I am curious if the authors have any insight into this.

Yes, we thought a lot about this peak in European emissions in 2021, but our ideas were all too vague to imply it in the discussions. An obvious suspicion would of course be, that it is related to the start of the Corona pandemic. Maybe maintenance of equipment and control mechanisms were reduced in that period, leading to more leakage. However, we could not find any more indication for this suspicion. Another idea was, that it is related to the emissions from soundproof windows. Especially in Western Europe, starting in 1975, $SF_6$ was filled in double-glazing windows to dampen acoustic pressure and improve the sound-insulating effect. In Germany, for instance, six percent of the manufactured and installed glazing contained $SF_6$ in 1990. Since the end of the 90's there was a transition away from the use of $SF_6$ for windows and the EU banned its use in 2008. The expected lifetime of soundproof windows is about 25 years and emissions of soundproof windows are still substantial in Germany (estimates are around 0.13 Gg/yr which is a huge fraction) and therefore also in Europe (since Germany is by far the biggest emitter in Europe and we see the same peak if we only look at German $SF_6$ emissions). Regarding this timeframe, it would fit that around 2020 the lifetime of a lot of windows that were built in the 90's come to an end which could lead to an emission increase followed by a decline. However, it is not so likely that a gradual replacement of such windows would cause a clear peak in a single year. The least exciting explanation would be, that this peak is a result of methodological uncertainties. Many inverse studies seem to show unrealistic interannual variability in posterior emissions, which could have its origin in the interannual variability of available observations, model errors, or a poor characterization of the emission and observation uncertainties. The 2020 peak is not particularly pronounced, so could well be an artifact.

Line 351: the citation here is the conference abstract version and I do not think they provide insight into whether there is any underestimation from Simmonds et al., neither in An et al. 2024. Perhaps just remove the citation here. As you claim that your emissions in China are more influenced by Gosan site, you can look further in to the reason for the difference between Simmonds and other emissions, rather than refer to a previous publication.

Yes, we removed the citation!

Line 359ff: I suggest that the discussion of emissions in these potentially less-constrained regions is accompanied by the discussion of error reductions. For GAINS prior, have you tried to increase the prior uncertainty and test it? You stated in the methods part that you did the relevant sensitivity tests with different prior emission uncertainties. I am afraid the inversion cannot constrain this region at all when using GAINS prior (no error reduction in Fig. 6).

Yes, for these regions the minimal a priori emission (and thus, also assumed uncertainty) is the driving variable. In some regions, e.g. Africa, we saw, that increasing the (minimal) emission uncertainty by a factor of 5 resulted in higher GAINS *a posteriori* emissions and a positive trend (see Figure below), which appears reasonable when comparing to the other inventory estimates. So, while the inversion is of course

limited in these regions, we think that results could give at least an indication of the direction in which emissions should be corrected.  We think the error reduction alone might give an incomplete picture here. It depends a lot on the estimated a priori emission uncertainty and on the a priori emission field itself. While the inversion very likely can not improve priors close to the "true" values in these regions, it is more likely to reduce large biases. Also, we see that the inversion leads to a positive trend for all tested a priori fields.

[Figure]

In other regions, e.g. Australia, also higher assumed prior uncertainties (5 times higher minimal a priori emission error - see Figure below) do not substantially change inversion results except leading to more inter-annual variability.

[Figure]

Line382: "see Sec. 3" in the bracket. please specify the specific section here for the prior uncertainty assignment.

done

Line 402ff: for AGAGE 12-box global SF6 emissions, you can refer to the latest ozone assessment report (providing emissions up to 2020) or An et al. 2024 (providing emissions up to 2021).

[Laube, J. C. & Tegtmeier, S. Chapter 1: Update on Ozone-depleting Substances (ODSs) and Other Gases of Interest to the Montreal Protocol. in *Scientific Assessment*

*of Ozone Depletion: 2022* vol. 278 (World Meteorological Organization, Geneva, Switzerland, 2022).

An, M. *et al.* Sustained growth of sulfur hexafluoride emissions in China inferred from atmospheric observations. *Nat Commun* **15**, 1997 (2024).]

Yes, thank you. We exchanged the extrapolated values with the updates from 2019-2021.

We changed the corresponding text to:

To judge the quality of our a posteriori global emission, we compare our results with the global emissions calculated by Simmonds et al. (2020) for the years 2005 to 2018 using the AGAGE 12-box model (e.g., Rigby et al., 2013), which we linearly extrapolated until 2021.  -> To judge the quality of our a posteriori global emission, we compare our results with the global emissions calculated by Simmonds et al. (2020) for the years 2005 to 2018 and updated until 2021 (An et al., 2024; Laube et al., 2023) using the AGAGE 12-box model (e.g., Rigby et al., 2013).

And some values in the comparisons slightly changed:

….while the UNFCCC-ELE \textit{a posteriori} global emissions are on average **16%** higher. -> ….while the UNFCCC-ELE \textit{a posteriori} global emissions are on average **18%** higher.

**….**emissions show on average almost no bias (**0.1%**) compared to the reference values -> emissions show on average almost no bias (**<1%**) compared to the reference values

The average of the total global emissions of the different discussed cases provides a very good estimate for the global SF6 emissions, showing an average bias of +**1,4%** compared to both, the AGAGE box model and the NOAA growth rate emissions ->

The average of the total global emissions of the different discussed cases provides a very good estimate for the global SF6 emissions, showing average biases of +**2,2% and 1.4%** compared to the AGAGE box model and the NOAA growth rate emissions

with average biases to the box model and NOAA growth rate emissions of **+16%, 0.1%, and -15%** for UNFCCC-ELE, EDGAR, and GAINS respectively

with average biases to the box model and NOAA growth rate emissions of +**18%, -15%, and <1%** for UNFCCC-ELE, GAINS and EDGAR respectively.

Line 418-419: I do not believe this could be the case. The ocean is very heterogeneous and the ocean flux is highly dependent on the locations (see Gruber et al. 2001). I suggest you discuss the potential uncertainties from that previous study you cite (Ni et al. (2023)) arising from scaling measured flux from a region (with potential strong

sink) to global (with strong sources at other regions). Your explanation in lines 425ff seems plausible. Also, in this paragraph, always clarify that the overestimation is specific to the UNFCCC-ELE inversion, not all the inversion in the study, to avoid confusion.

[Gruber, N., Gloor, M., Fan, S.M. and Sarmiento, J.L., 2001. Air-sea flux of oxygen estimated from bulk data: Implications for the marine and atmospheric oxygen cycles. *Global Biogeochemical Cycles*, *15*(4), pp.783-803.]

We agree that the estimates of Ni et al. are probably highly uncertain and added:

They estimated this global oceanic sink by scaling up calculations of sea-air fluxes based on simultaneous measurements of $SF_6$ concentrations in the atmosphere and surface seawater of the Western Pacific and Eastern Indian Ocean. However, since the ocean fluxes are highly inhomogeneous  (strong oceanic sources might exist in other regions), we suspect the up-scaled estimate to be very uncertain. Nevertheless, we ......

increase of the global emissions by the inversion -> increase of the global **UNFCCC-ELE** emissions by the inversion

There is a better explanation for our too-high \textit{a posteriori} emissions. -> There is a better explanation for our too-high \textit{a posteriori} **UNFCCC-ELE** emissions.

L440: "underestimation of the emission residuals between the global and the Chinese emissions", clarify that it is in GAINS prior emissions.

done

L454-456: consider the uncertainties for the trend for both global and China.

We changed:

Notice that the average global trend of 0.20\;Gg/yr is slightly smaller than for Chinese emissions (0.21  Gg/yr), supporting the finding of An et al. (2024) that Chinese emissions alone have offset the overall decreasing emissions from all other countries --> Notice also, that the average global trend (0.20 Gg/yr) is similar to the Chinese emission trend (0.21 Gg/yr),

L457-458: state here that your bias may be especially in the poorly-observed regions.

We added:

Despite some potential problems with our inversion setup that can lead to biased a posteriori global emissions (as could be clearly seen and explained with the UNFCCC-ELE and GAINS a priori emissions in poorly-observed regions )

Line 488-490: is there any result to support your statement here that your results are mainly driven by the high-frequency data in the U.S.? In addition, you can also have a look at the mole fraction enhancements, either in the observations or the posterior simulated ones, to check the seasonality in mole fraction enhancements. Do you have any data with reference (e.g., the high power transmission in summer in Line 490-491) to help justify your seasonal cycle?

No, you are right, this was more of an assumption.

We rephrased:

In addition, our inversion results for the USA are mainly driven by the high-frequency measurements from Trinidad Head (THD) and Niwot Ridge (NWR), which have not been used by Hu et al. (2023) -> In addition, high-frequency measurements from Trinidad Head (THD) and Niwot Ridge (NWR), have not been used by Hu et al. (2023)

Yes, thank you, we actually looked at observed mole fraction enhancements at background stations, however, those analyses didn't provide us with a clear picture.

Yes, for instance, data from the U.S. Energy Information Administration. We added:

(see e.g. https://www.eia.gov/electricity/).

Line 520ff: Again, you need to consider the uncertainties. If the two trends are not significantly different, then I suggest you remove this bit in the conclusion. In addition, I suggest also mentioning that the China's official voluntary reports are improved in the latest reports compared to the top-down results (Figure 10), and also discussing this in the main text.

We added:

The values from the more recent reports in 2017 and 2018 are, however, closer to our inversion results, indicating an improvement in Chinese reports.

We removed:
The derived trend is slightly steeper than the global total SF$_6$ emission trend (0.20 Gg/yr), supporting the suggestion that Chinese emissions alone have more than offset the overall decreasing emissions from other countries \citep{Minde_an_2024}

We added: however, the latest reports in 2017 and 2018 seem to be improved.

References:

Lunt, M. F., Manning, A. J., Allen, G., Arnold, T., Bauguitte, S. J.-B., Boesch, H., Ganesan, A. L., Grant, A., Helfter, C., Nemitz, E., O'Doherty, S. J., Palmer, P. I., Pitt, J. R., Rennick, C., Say, D., Stanley, K. M., Stavert, A. R., Young, D., and Rigby, M.: Atmospheric observations consistent with reported decline in the UK's methane emissions (2013–2020), Atmos. Chem. Phys., 21, 16257–16276, https://doi.org/10.5194/acp-21-16257-2021, 2021.

---

## Author Comment (AC2)

**Author's response #2**

Regional and global atmospheric observation-based estimates of SF6 emissions are presented using a Lagrangian inverse modelling system. Emissions trends are derived for the major emitting regions, China, the USA and EU, which are generally consistent with other available regional studies, but mostly higher than reported emissions. Global emissions are also broadly consistent with previous studies.

The article is detailed and meticulous and very well written. The methods are interesting and novel, and the application is important and timely. I think the paper is suitable for publication in Atmospheric Chemistry and Physics, subject to some minor corrections.

We would like to thank reviewer #2 for the detailed and very productive review of our manuscript.

In the response we use 4 different colors. The blue colored text is the general answer to the reviewer's comments. Additionally, we show how the text is changed in the manuscript: The original text is colored grey, removed text is colored red, and new text is colored green.

Main text:

L15: Point 5 in this list is somewhat confusingly worded. Perhaps something like: "Global total SF6 emissions are comparable to previous studies but are sensitive to a priori estimates, because of the poor network sensitivity to some regions (e.g., Africa, South America)""

 We changed this, following largely the reviewer's suggestion:

The global total SF6 emissions are captured well by the inversion, however, results are sensitive to the a priori emission estimates, given that substantial biases of these estimates in regions poorly covered by the measurement network (e.g. Africa, South America) can be improved but not entirely corrected. -> Global total $SF_6$ emissions are comparable to estimates in previous studies but are sensitive to a priori estimates, due to the low network sensitivity in poorly monitored regions.

L22 and L27-29: I suggest deleting the lines beginning "However, this GWP 100 value…" and "Thus, GWPs, which are typically…". I don't think it's accurate to say that the GWP 100 value "underplays the climate impact of this gas". If you wanted to examine the climate impact over longer timescales, you could define a longer-term GWP. It's well known that GWP has several flaws, but I don't think you need to go into them here.

Yes -> done

L43: I'd separate out the part of this sentence on SF6 measurements being used to determine stratospheric OH into its own sentence. The other parts of this list are sources, whereas this is a measurement of atmospheric SF6. You could also add that it has been used as an ocean tracer though.

done -> changed it to:

Furthermore, SF6 finds applications in semiconductor manufacturing, facilitating precise etching processes (Lee et al., 2004) and serves for blanketing or degassing in the magnesium or aluminum metal industry (Maiss and Brenninkmeijer, 1998). Moreover, it is used in medicine (Lee et al., 2017; Brinton and Wilkinson, 2009), photovoltaic manufacturing (Andersen et al., 2014), military applications (Koch, 2004), particle accelerators (Lichter et al., 2023), soundproof glazing (Schwarz, 2005), sports shoes (Pedersen, 2000), car tyres  Schwaab, 2000), wind turbines (EPA, 2023) and as a tracer gas in the atmosphere (Martin et al., 2011), in groundwater (Okofo et al., 2022), rivers (Ho et al., 2002), and oceans (Tanhua et al., 2004).

L44 and 54: I suggest removing "developed" and "developing". These terms are not needed here.

done

L70: This statement isn't true, as Rigby et al., 2011 was a global inverse modelling study that used a 3D (Eulerian) model.

Yes, we agree and changed this to:

Up to this point, SF6 inversion studies have exclusively been focusing on specific geographical areas, i.e., using regional inversions only. Although global observation-based box models, such as the AGAGE 12-box model (e.g., Rigby et al., 2013) are considered to be capable of accurately determining the global total emissions, a comprehensive top-down perspective of the global SF6 emission distribution is missing. -> Although global SF6 emissions can be well constrained by global box models, such as the AGAGE 12-box model (e.g., Rigby et al., 2013), and regional inversion systems have been used to estimate SF6 emissions in specific regions, there is no clear link between regional and global emissions and an updated, comprehensive top-down perspective of the global SF6 emission distribution is missing.

L113: Measurement location and time?

Yes - done

L115 – 118 and throughout the following sections: I think you need to be careful with the notation here. In this section, where you define He, e, etc. it implies that these sensitivities are to the grid-scale emissions or mole fraction fields. However, you've

used a basis function decomposition of the emissions field in your inversion (but I'm not sure how you're scaling your initial conditions field, see below). Therefore, the matrices and vectors in Equations 2 and 3 are different to those defined here. I think you could make this consistent by stating that e, He, etc are for aggregated groups of grid cells when you define them?

Yes, thank you, that is true!

We added: Note at this point, that we aggregate grid cells of the emission grid for the optimization (see Sec.2.5) and that the just-defined variables (He, e, Hi, yi) refer to aggregated groups of grid cells. For a detailed description please see (Thompson and Stohl (2014).

Figure 2: How have you dealt with the different frequency between the flask and high-frequency data here? Is this the average over all time points, with zeros during times where there are no flask data?

The average is the sum of all sensitivity fields (representing the sensitivity to one observation respectively) divided by the total number of sensitivity fields. Thus, there is a weighting of the sensitivity of different measurement sites according to the measurement frequency. To clarify this, we have added to the figure caption: "Notice that values represent averages over all cases, for which FLEXPART calculations were made. Thus, sites with high-frequency on-line observations are weighted more strongly than sites where only flask measurements are made, or observations from moving platforms."

Section 2.5: Please clarify:

- if emissions and boundary conditions are being *scaled* in the inversion, or if absolute values are being derived. Furthermore, how are grid cells aggregated within the spatial basis functions? Is the spatial pattern of the underlying grid cells preserved, or are emissions spread out uniformly within the aggregated cells?

Absolute values of emissions are derived, while the boundary conditions are scaled. Emissions in the fine grid are weighted by the ratio of the area of the fine grid to the variable grid, into which it is aggregated. After the inversion, optimized emissions in the variable-resolution coarse grid were redistributed onto the fine grid according to the relative distribution of the a priori emissions.

We added: "Emissions in the fine grid are thereby weighted according to the ratio of the area of the fine grid to the variable-resolution coarse grid into which it is aggregated. After the inversion, optimized emissions in the variable grid were redistributed onto the fine grid according to the relative distribution of the a priori emissions."

- how the initial conditions are being adjusted. Is the whole field adjusted each month (or, equivalently, are the baseline mole fractions at the stations being adjusted uniformly? Or perhaps adjusted on a per-station basis?), or is there some spatial decomposition?

Yes, the whole field is adjusted every month.

We added: ... , where the whole field is adjusted on a monthly basis.

- Does R contain only "observational errors", as stated? If so, how is this defined (i.e., is it just measurement repeatability)? And if this is the case, what about model (or mismatch) uncertainty? How have you accounted for this critical (but highly uncertain) term? It seems that this term should also be the subject of a sensitivity test.

Yes, we accounted for the model uncertainty.

We added: "FLEXINVERT+ assumes a diagonal observation error covariance matrix R, and therefore, does not account for possible error correlations between different observations. The diagonal elements represent the sum of measurement and model error, where we assume the latter to be dominant. Our error estimates are based on a number of initial inversion runs, where we assessed the model error according to the a posteriori model residuals (difference between observed and a posteriori simulated mole fractions), and such that the reduced chi-square value (the value of the cost function at minimum divided by the number of observations and divided by 2) is close to 1."

- Have the observational data been filtered at all? For example, excluding points under low boundary layer heights, or at night, as is often done due to poorer model performance under these conditions? Furthermore, note that SF6 mole fractions in populated regions show occasional very large events, perhaps linked to equipment failure (see, for example, the note that very large emissions are derived during some months, here: https://assets.publishing.service.gov.uk/media/62d7b9bee90e071e7e59c97 e/verification-uk-greenhouse-gas-emissions-using-atmospheric-observations-annual-report-2021.pdf). Do these need to be excluded, since your emissions model assumes constant fluxes (at least during each month)?

Yes, we excluded occasional very large events.

We added: "In addition, we adopted a method by Stohl et al. (2009) to identify observations that cannot be brought into agreement with modeled mole fractions by the inversion, which we removed entirely (in contrast to Stohl et al., 2009, who assigned larger uncertainties to these observations). For this, we utilized the kurtosis of the a posteriori error frequency distribution and iteratively excluded observations causing the largest absolute errors until the

kurtosis of the remaining error values fell below 5, approximating a Gaussian distribution)."

Otherwise, we did no filtering. See comment to reviewer 1.

- How was the baseline uncertainty of 0.15 ppt, and correlation length scales, arrived at? Why 70% for the prior uncertainty?

These values are based on sensitivity tests and values previously used in the literature, but are of course debatable. The emission data, unfortunately, do not contain uncertainty information, such that any value used is ambiguous and requires subjective judgement. The results are not very sensitive to the choice of the baseline uncertainty.

- I don't understand why a 70% level of prior uncertainty on a per-grid cell basis doesn't lead to a vanishingly small prior global uncertainty. Can you clarify? If you have ~5000 grid cells, wouldn't the global uncertainty be ~70% / sqrt(5000), which is ~1% (notwithstanding spatial correlations and minimum values).

The reviewer is right. We revised the uncertainties, which for global emissions indeed become very small. We therefore base our uncertainty estimates also on the differences obtained when using different a priori inventories.

- Surely the temporal correlation of 90 days plays very little role, given that you are solving for annual emissions in the main results? Is this term needed?

Yes, you are right. The term plays little role and is probably not needed in this case, even though it helps to regularize the problem.

L253 – L256: I would remove these statements (or at least the sentence on L256), as it suggests the inversion has more capacity to focus on "incorrect" parts of the model than it really has. It is of course better if the prior model baseline is better, but the optimization is of the whole system. Even if the prior model simulated a perfect baseline, errors in sensitivities to boundary conditions or footprints could still lead to an adjustment away from that perfect baseline.

We have removed the sentence: "This is important, as the optimization can focus on improving the emissions rather than correcting a wrong baseline". We left the rest of the text, since we think it is relevant to point towards the importance of a well-fitting baseline.

L261 – 265: I think these lines should be removed. I don't doubt that a 50-day simulation period is more "accurate" than a 10-day period. But it's not shown here.

We removed:

"Figure 5b further illustrates the advantage of choosing a rather long 50-day backward simulation period. With this long simulation period, we can see that this remote station is also directly influenced by emissions (i.e., enhancements over the baseline) that can be directly optimized. With shorter simulation times (e.g., 5-10 days), no emission contributions above the baseline could be seen, thus rendering this station useless for emission optimization. For a detailed discussion about the LPDM backward simulation period see Vojta et al. 2022."

L289 and throughout this section. Please provide an uncertainty to these quantities.

Yes -> done

L298 – 299: Remove the sentence about it being "reassuring". This is subjective and not needed.

Yes -> done

Section 3.3.4: My reading of all of these subsections is basically that there is, not surprisingly, very sensitivity to these regions. I suggest moving this content to the Supplement and summarizing this message in a paragraph or two in the main paper.

Thank you for the suggestion, but in this case, we would like to keep it as it is. Firstly, we would like to illustrate the big differences of the different a priori inventories in those regions, which are rarely discussed elsewhere. Also, (without overestimating the inversion's capability in those regions), we still think it is interesting that the results in many cases at least indicate a positive trend (even if very uncertain) and that the inversion derives smaller posterior emissions for the UNFCCC-ELE inventory compared to the prior which we suspect to overestimate emission in those regions (due to the overestimation of the global a posteriori emissions).

L434: Should this be "is larger than, and inconsistent with, the global atmospheric SF6 growth…". Furthermore, I wouldn't use "postulated" in this sentence (use "derived" or similar).

Yes -> done

L461: "could be brought relatively close to these previous estimates", rather than "known values" (there are no "known values").

Yes -> done

L462: Suggest deleting "which has rarely been achieved before", as it's too broad here. There are many studies using global Eulerian models that do this (although only one for SF6 that I'm aware of; i.e., Rigby et al., 2011).

Yes -> done

L462 – 468: I don't agree with the framing of these sentences. The novelty of this work is that it attempts to create a global picture using a backward running Lagrangian model. This is very nice in itself. But we shouldn't get carried away that 50-day back trajectories can really give us a full global picture, given the sparse measurement network. As this work shows, there is negligible sensitivity to large parts of the world, irrespective of the integration time. Without additional measurements, emissions derived from these regions will always be subject to biases from the prior and the accumulation of transport errors. Furthermore, the last part of these sentences is conjecture, that there is a "clear direction" in the adjustments to these unsampled regions. This seems to be subjective to me. I suggest cutting these sentences. The work is impressive in itself. You don't need to over-sell it.

 Agreed.

-> We deleted: We attribute this capability of simultaneously constraining both regional as well as global emissions mostly to our long backward calculation period of 50 days (Vojta et al. 2022) and our extensive observation data set.

-> and : Nevertheless, in most cases, the regional results at least indicate a clear direction in which \textit{a priori} emissions need to be corrected even for these poorly monitored regions.

Section 3.3.6: Note that seasonal emissions were also briefly noted for north-east Europe in Reddington et al. (2019). Similarly to Hu et al, these maximized in the winter.

Thank you for this reference, which we have added to the paper. Indeed, they mention a winter maximum and their Figure 111 seems to suggest it; however, without showing a very systematic seasonal cycle.

We rewrote:
While there is no clear seasonal cycle in the EU emissions, the Chinese seasonality is similar to the one in the Northern Hemisphere (Fig. 13b) ->

For EU emissions no clear seasonal cycle can be seen. Notice at this point that $SF_6$ emissions from North-West Europe were found to maximize in the winter  (Redington et al., 2019) however without showing a very systematic seasonal cycle. For Chinese $SF_6$ emissions, the seasonality is similar to the one in the Northern Hemisphere (Fig. 13b).

L499: I suggest "boundary conditions", rather than "initial conditions"

Yes -> done

L502 – 503: I suggest deleting the final sentence for the reasons outlined above (comment to L253)

Yes -> done

L504 – 505: I also suggest deleting the final sentence of this bullet for the reasons outlined above (comment on L462)

Yes -> done

L509 and throughout this section: provide uncertainties

Yes -> done

L517: Delete the final sentence, as this is conjecture.

 We changed the sentence to:

This might suggest that the EU's new F-gas regulation was almost immediately successful in reducing SF6 emissions.

L527: Delete the two final sentences, as I don't see how you could know this. It's not supported by your investigation.

Our results showed that, when using the EDGAR prior emissions, the global a posteriori emissions showed the best agreement to the total global reference values. We therefore think that the aggregated prior estimated emission in poorly covered regions (residual between global emissions and emissions in well-monitored areas), should be a relatively good estimate, as otherwise we would expect larger biases in the global emissions.

We changed the two sentences to: The EDGAR bottom-up inventory seems to provide a relatively good estimate for the total emissions aggregated over all the poorly monitored regions (residual between global emissions and emissions in well-monitored areas), as otherwise, the global a posteriori emissions would be more strongly biased against the relatively well known global emissions based on atmospheric growth rates. Nevertheless, more observations are needed to investigate if also regional emission patterns in those areas are accurate.

L529 – 530: I think this bullet should be deleted, as there's so little sensitivity to this region.

We rephrased the bullet point:

Our inversions suggest globally significant and strongly increasing emissions in India since 2005. However, the results for this region are very uncertain because of a weak observational constraint. Adding monitoring capacity in this region should be a high future priority.

L542: Delete the final bullet, as it's well outside the scope of your work.

Yes -> done

Supplement:

L14: full stop needed.

Yes -> done

L44: Please confirm that the following is correct and has been checked in your analysis: The cited paper (Guillevic et al., 2018) quotes the ratio NOAA-2014 / SIO-05 = 1.002 ± 0.002. However, the wording on this line suggests that conversion from NOAA-14 to SIO-05 is by multiplication by 1.002. The cited reference suggests that division by 1.002 would be required.

 Yes, we divided by the factor 1.002 and rephrased:

We used the factor -> we divided by the factor

References

Redington, A. L., Manning, A. J., O'Doherty, S. J., Say, D., Rigby, M., Hoare, D., Wisher, A., Rennick, C., Arnold, T., Young, D., and Simmonds, P. G.: Long-Term Atmospheric Measurement and Interpretation of Radiatively Active Trace Gases, Annual Report, Sept 2018 – Sept 2019, Department for Business, Energy and Industrial Strategy, London, UK, https://assets.publishing.service.gov.uk/media/5eddf868d3bf7f4601e57730/verification-uk-greenhouse-gas-emissions-atmospheric-observations-annual-report-2018.pdf, 2019.

---

## Author Response (AR2)

**Author's response**

I don't think the authors have addressed my comment that this is not the first non-box model global study. They have now added the line: "Although global SF6 emissions can be well constrained by global box models, such as the AGAGE 12-box model (e.g., Rigby et al., 2013), and regional inversion systems have been used to estimate SF6 emissions in specific regions, there is no clear link between regional and global emissions and an updated, comprehensive top-down perspective of the global SF6 emission distribution is missing." However, this line still omits the previous global studies on SF6 (Rigby et al., 2010; 2011), but furthermore, I don't really know what it means to say that there is "no clear link" between the regional and global scales. I suggest citing the previous global studies in the first part of this sentence, removing the middle part, and saying that an update is needed.

We changed:

Although global SF6 emissions can be well constrained by global box models, such as the AGAGE 12-box model (e.g., Rigby et al., 2013), and regional inversion systems have been used to estimate SF6 emissions in specific regions, there is no clear link between regional and global emissions and an updated, comprehensive top-down perspective of the global SF6 emission distribution is missing. ->

Global total SF6 emissions can be well constrained by global box models, such as the AGAGE 12-box model (e.g., Rigby et al., 2013). While Rigby et al. (2010, 2011) presented global SF6 inversion studies and recent regional studies have estimated SF6 emissions in specific regions (e.g. Hu et al. 2024, An et al. 2024), an updated, comprehensive top-down perspective of the global SF6 emission distribution is needed.

In response to my previous question about the a priori uncertainty, when aggregated to the global scale, the authors say "We therefore base our uncertainty estimates also on the differences obtained when using different a priori inventories." I didn't realise this from the previous version of the paper and see that this has now been explained in the text of Section 3. However, this is important for readers to understand, so I also think it needs to be justified in Section 2, and included in the figure captions (which currently imply that it's the a posteriori uncertainty). Furthermore, I think it's misleading to describe these uncertainties as "1-sigma" in the figure captions and elsewhere, as this would imply that they represent one standard deviation over some PDF. But in reality, they are no longer tied to some known distribution. They are a somewhat arbitrary measure of the sensitivity of the results to the priors.

Agreed, we added to Sec 2.6:

In order to reflect the sensitivity of the results to the *a priori* emissions we define the uncertainty intervals of aggregated emissions as the minimum and maximum 1-σ uncertainty limits across the inversion results using the different *a priori* emissions.

In the figure captions, we changed:

*A posteriori* emissions are illustrated together with their respective 1-σ uncertainties (colored shadings). -> *A posteriori* emissions are illustrated together with their respective uncertainties (colored shadings, defined as the minimum and maximum 1-σ uncertainty limits across the inversion results for different *a priori* variations).